# Discovery of non-squalene triterpenes

Hui Tao[1,9], Lukas Lauterbach[2,9], Guangkai Bian[3,9], Rong Chen[4,9], Anwei Hou[2,9], Takahiro Mori[1,5,6,9], Shu Cheng[7], Ben Hu[7], Li Lu[7], Xin Mu[7], Min Li[7], Naruhiko Adachi[8], Masato Kawasaki[8], Toshio Moriya[8], Toshiya Senda[8], Xinghuan Wang[4], Zixin Deng[7], Ikuro Abe[1,5✉], Jeroen S. Dickschat[2✉] & Tiangang Liu[3,4,7✉]

All known triterpenes are generated by triterpene synthases (TrTSs) from squalene or oxidosqualene[1]. This approach is fundamentally different from the biosynthesis of short-chain ($C_{10}$–$C_{25}$) terpenes that are formed from polyisoprenyl diphosphates[2–4]. In this study, two fungal chimeric class I TrTSs, *Talaromyces verruculosus* talaropentaene synthase (TvTS) and *Macrophomina phaseolina* macrophomene synthase (MpMS), were characterized. Both enzymes use dimethylallyl diphosphate and isopentenyl diphosphate or hexaprenyl diphosphate as substrates, representing the first examples, to our knowledge, of non-squalene-dependent triterpene biosynthesis. The cyclization mechanisms of TvTS and MpMS and the absolute configurations of their products were investigated in isotopic labelling experiments. Structural analyses of the terpene cyclase domain of TvTS and full-length MpMS provide detailed insights into their catalytic mechanisms. An AlphaFold2-based screening platform was developed to mine a third TrTS, *Colletotrichum gloeosporioides* colleterpenol synthase (CgCS). Our findings identify a new enzymatic mechanism for the biosynthesis of triterpenes and enhance understanding of terpene biosynthesis in nature.

Triterpenoids are a large class of natural products with a broad range of bioactivities[5] known from all kingdoms of life[6,7]. Their biosynthesis starts with dimerization of farnesyl diphosphate (FPP) by squalene synthase into squalene[8]. The cyclization is catalysed by TrTSs such as squalene hopene cyclase (SHC) to yield pentacyclic hopene, the precursor of hopanoids[9]. After oxidation to (*S*)-2,3-epoxysqualene, lanosterol synthase (LS) promotes the biosynthesis of lanosterol, the precursor to steroids and saponins[10] (Fig. 1). These enzymes are classified as class II terpene synthases (TSs) with a conserved DXDD motif for substrate activation through protonation. By contrast, class I TSs have two conserved DDXX(D/E) and NSE/DTE motifs for binding of a ($Mg^{2+}$)$_3$ cluster that in turn binds the substrate's diphosphate[11]. In cooperation with the diphosphate sensor, a conserved arginine forming hydrogen bridges to the diphosphate, and an effector residue at a helix break[12] the Lewis acidity of $Mg^{2+}$ leads to substrate ionization through diphosphate abstraction. Many examples are known of class I TSs[13] that convert substrates with chain lengths between $C_{10}$ and $C_{25}$ into usually (poly)cyclic terpenes. By contrast, class I cyclization of hexaprenyl diphosphate (HexPP; $C_{30}$) into triterpenes is unknown, although HexPP is widely distributed in nature[14–16] and serves as a precursor for ubiquinone ($Q_6$) in yeast and for highly bioactive meroterpenoids in sponges[17]. Bifunctional enzymes harbouring a prenyl transferase (PT) and terpene cyclase (TC) domain (PTTCs) have recently attracted attention[3,18,19]. Their PT domains convert dimethylallyl diphosphate (DMAPP) and isopentenyl diphosphate (IPP) into polyisoprenyl diphosphates, and the TC domains carry out cyclization into di- and sesterterpenes. Thus far, no such enzyme is known to produce triterpenes. Here we describe the discovery of chimeric fungal class I TrTSs for the production and conversion of HexPP into triterpenes.

Using our yeast-based genome mining platform[20], two putative bifunctional TSs with PT and TC domains emerged from the endophyte *T. verruculosus* TS63-9 (ref. [21]; ZTR_06220) and the plant pathogen *M. phaseolina* MS6 (ref. [22]; MPH_02178) (Supplementary Figs. 1 and 2). These enzymes showed 40.4% and 33.4% amino acid sequence identity with the *Phomopsis amygdali* fusicoccadiene synthase[23] (PaFS) and the *Aspergillus clavatus* ophiobolin F synthase[24] (AcOS), respectively, and are located within clades II-B and II-C of fungal bifunctional TSs (Supplementary Fig. 3). Thus, both enzymes are predicted to follow a C1-III-IV cyclization mode with attack of the third double bond at C1 followed by attack of the fourth double bond[25].

Expression of ZTR_06220 in *Saccharomyces cerevisiae* YZL141 (ref. [20]) yielded the triterpene hydrocarbon talaropentaene (**1**) (Supplementary Table 3, Fig. 2a, d and Supplementary Figs. 4–12), characterizing the enzyme as TvTS. Transcriptional analysis showed that TvTS was not expressed in its native host, but placing the gene under the control of the *amyB* promoter resulted in the production of **1** and confirmed its natural function (Fig. 2b, d). Because recombinant full-length TvTS was insoluble, for in vitro assays the TvTS-PT and TvTS-TC domains were expressed individually (Supplementary Fig. 14). Incubation of TvTS-PT with DMAPP and IPP yielded HexPP (Fig. 2c), which was converted into

[1]Graduate School of Pharmaceutical Sciences, The University of Tokyo, Tokyo, Japan. [2]Kekulé-Institut für Organische Chemie und Biochemie, Rheinische Friedrich-Wilhelms-Universität Bonn, Bonn, Germany. [3]Department of Urology, Zhongnan Hospital of Wuhan University, School of Pharmaceutical Sciences, Wuhan University, Wuhan, China. [4]Department of Urology, Zhongnan Hospital of Wuhan University, Wuhan, China. [5]Collaborative Research Institute for Innovative Microbiology, The University of Tokyo, Tokyo, Japan. [6]PRESTO, Japan Science and Technology Agency, Saitama, Japan. [7]Key Laboratory of Combinatorial Biosynthesis and Drug Discovery, Ministry of Education and School of Pharmaceutical Sciences, Wuhan University, Wuhan, China. [8]Structural Biology Research Center, Institute of Materials Structure Science, High Energy Accelerator Research Organization (KEK), Tsukuba, Japan. [9]These authors contributed equally: Hui Tao, Lukas Lauterbach, Guangkai Bian, Rong Chen, Anwei Hou, Takahiro Mori. ✉e-mail: abei@mol.f.u-tokyo.ac.jp; dickschat@uni-bonn.de; liutg@whu.edu.cn

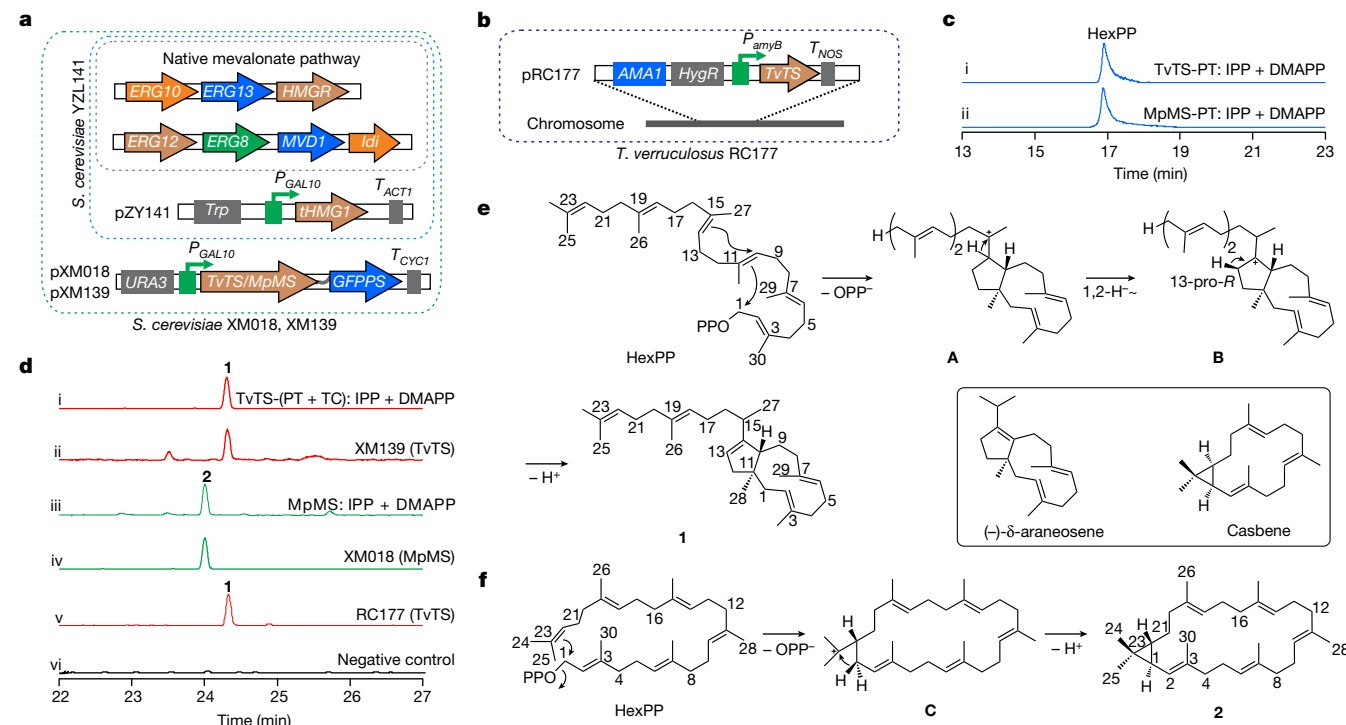

**Fig. 1 | Triterpene biosynthesis.** The classical pathway for triterpenes proceeds through squalene. This study describes three fungal bifunctional TSs that convert DMAPP and IPP through HexPP into the triterpenes talaropentaene (**1**), macrophomene (**2**) and colleterpenol (**3**). FPPS, FPP synthase; HexPPS, hexaprenyl diphosphate synthase; SQE, squalene epoxidase; SQS, squalene synthase.

**1** by TvTS-TC (Fig. 2d). The combination of TvTS-PT and TvTS-TC also accepted FPP and IPP, while TvTS-TC alone was sufficient to cyclize HexPP (Supplementary Scheme 1) to **1** (Supplementary Fig. 15).

The absolute configuration of **1** was determined through stereoselective deuteration (Supplementary Table 5)[26]. Conversion of geranyl diphosphate (GPP) and (*E*)- or (*Z*)-(4-13C,4-2H)IPP[27] with TvTS-PT and TvTS-TC indicated an absolute configuration of (10*R*,11*S*)-**1** (Supplementary Fig. 16). Similar experiments using GPP and (*R*)- or (*S*)-(1-13C,1-2H)IPP[28] with TvTS-PT and TvTS-TC confirmed this assignment for **1** (Supplementary Fig. 17). Compound **1** is structurally similar to (−)-δ-araneosene from the fungal clade II-D TS *C. gloeosporioides* dolasta-1(15),8-diene synthase (CgDS), including corresponding absolute configurations (Fig. 2e)[25].

**Fig. 2 | Characterization of TvTS and MpMS. a**, Engineering of *S. cerevisiae* for production of **1** (TvTS) and **2** (MpMS). **b**, Construction of *T. verruculosus* RC177. **c**, Ion chromatograms from high-resolution electrospray ionization mass spectrometry (EI-MS; *m/z* 505.3452) of extracts from incubation of DMAPP and IPP with (i) TvTS-PT and (ii) MpMS-PT D114A/N115A. **d**, EI-MS ion chromatograms of extracts from (i) incubation of DMAPP and IPP with TvTS-PT and TvTS-TC, (ii) *S. cerevisiae* XM139 expressing the gene for TvTS, (iii) incubation of DMAPP and IPP with MpMS, (iv) *S. cerevisiae* XM018 expressing the gene for MpMS, (v) engineered *T. verruculosus* RC177 with the TvTS gene under the control of the *amyB* promoter and (vi) wild-type *T. verruculosus* TS63-9 (negative control; Supplementary Fig. 13). **e, f**, HexPP cyclization to **1** (**e**) and **2** (**f**) with carbon numbering as in HexPP. Box, structurally related diterpenes.

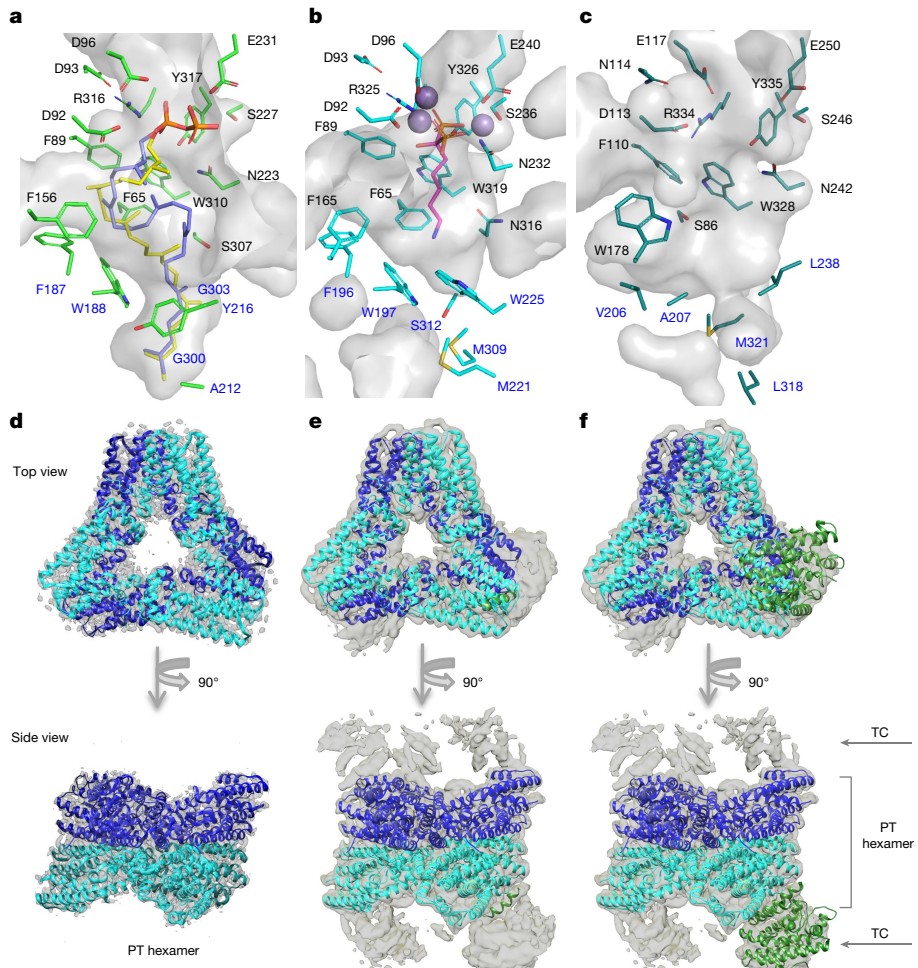

**Fig. 3 | Structures of TvTS and MpMS. a–c,** Active sites in the docking model of TvTS-TC with 2,3-dihydro-HexPP (two possible conformers were docked on the basis of the observed electron density and are shown as yellow and purple sticks) (**a**), PaFS-TC (**b**) and MpMS-TC (**c**) (modelled by AlphaFold2). **d,** Cryo-EM map and reconstructed structure of non-cross-linked hexameric MpMS-PT (monomers in blue and cyan; map resolution of 3.17 Å; density map contoured at 0.065 using Chimera). **e,** Cryo-EM map and reconstructed structure of cross-linked MpMS-PT hexamer (map resolution of 4.00 Å; density map contoured at 0.030 using Chimera). The cross-linked TC domain helix is shown in green. **f,** Reconstructed structure of the MpMS-PT hexamer with a TC domain homology model (based on FgGS; Protein Data Bank (PDB), 6W26) docked into the cryo-EM map.

The proposed cyclization mechanism for **1** starts with diphosphate abstraction from HexPP, followed by C1-III-IV cyclization to **A**. A subsequent 1,2-hydride shift to **B** and deprotonation generate **1** (Fig. 2e). Experimental proof for the 1,2-hydride shift was obtained through incubation of $(3-^{13}C,2-^{2}H)FPP^{29}$ and IPP with TvTS-PT and TvTS-TC, yielding $(15-^{13}C,15-^{2}H)$-**1**. This product showed an intensive upfield-shifted triplet ($\Delta\delta = -0.45$ ppm, $^{1}J_{C,D} = 18.9$ Hz) for C15 indicating a direct $^{13}C–^{2}H$ bond (Supplementary Fig. 18). Incubation of GPP and $(R)$- or $(S)$-$(1-^{13}C,1-^{2}H)$ IPP with TvTS-PT and TvTS-TC resulted in specific loss of the 13-pro-$R$ hydrogen of **B** in the deprotonation to **1** (Supplementary Fig. 19).

Expression of MPH_02178 in *S. cerevisiae* resulted in macrophomene (**2**), albeit with low yield, but DMAPP and IPP were efficiently converted into the same triterpene using purified recombinant enzyme (Supplementary Figs. 20–22). Consequently, the TrTS was identified as MpMS. Analysis of an MpMS D114A/N115A variant with an inactivated TC domain demonstrated that HexPP was produced by the PT domain (Fig. 2c). In addition, GPP, FPP, geranylgeranyl diphosphate (GGPP) and geranylfarnesyl diphosphate (GFPP) in combination with IPP and synthetic HexPP were also accepted by MpMS to yield **2**. For structure elucidation of **2** (Supplementary Table 6 and Supplementary Figs. 23–30), the MpMS E104Y variant was selected[30] (Supplementary Fig. 31). To enable full assignment of the nuclear magnetic resonance (NMR)

data, which was prevented by multiple peak overlaps for unlabelled **2**, isotopic labelling experiments with $^{13}C$-labelled precursors were performed (Supplementary Fig. 32a–e).

Notably, **2** is structurally similar to casbene. Both compounds are bicyclic with one macrocyclic ring and one three-membered ring, except the ring fusion is *trans* for **2** and *cis* for casbene (Fig. 2f)[31]. To our knowledge, the 22-membered ring in macrophomene represents the largest macrocycle discovered in terpenes so far and is only reflected by 22-membered rings in nostocyclophanes, a class of oxidatively dimerized polyketides from cyanobacteria[32]. The proposed cyclization mechanism from HexPP to **2** starts with diphosphate abstraction and 1,22-cyclization to **C**, followed by deprotonation at C1 to close the cyclopropane ring. This mechanism represents an unprecedented C1-VI cyclization mode (Supplementary Fig. 3, designated type C) for chimeric TSs. The stereochemical fate of the geminal methyl groups at C24 and C25 of HexPP was addressed by incubating $(12-^{13}C)FPP^{33}$ and $(9-^{13}C)GPP^{25}$ with IPP and MpMS. Product analysis by $^{13}C$-NMR showed a clear stereochemical course for the attack at C23 in cyclopropane ring closure (Supplementary Fig. 33). The stereochemistry of the deprotonation was addressed by incubation with GFPP and $(R)$-$(1-^{2}H)$ IPP or $(S)$-$(1-^{2}H)IPP^{34}$, with deuterium retained for the $(R)$ enantiomer and lost from the $(S)$ enantiomer (Supplementary Fig. 34). Taken

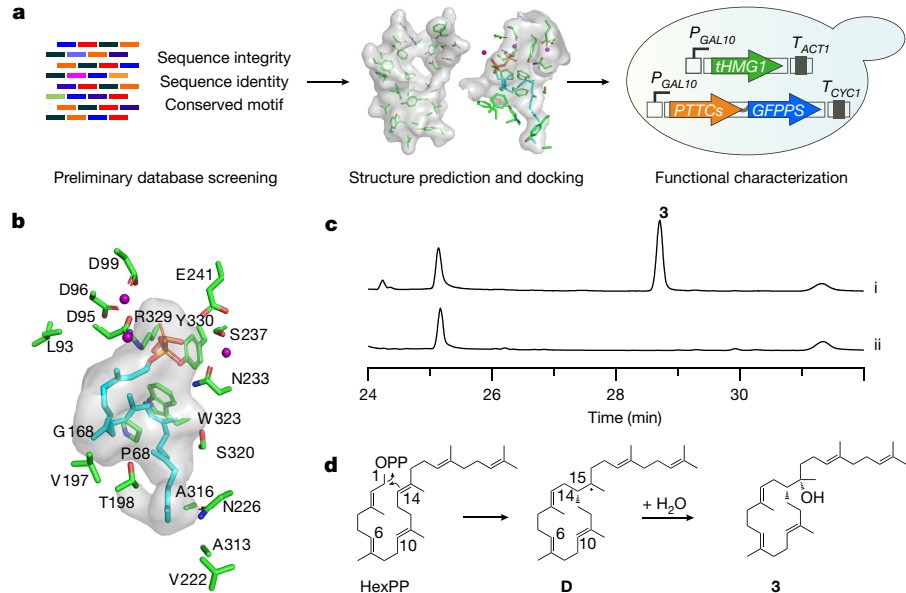

**Fig. 4 | AlphaFold2-based genome mining of CgCS. a**, AlphaFold2-based screening of chimeric class I TSs. **b**, Predicted structure of CgCS-TC docked with HexPP (purple spheres, $Mg^{2+}$). In CgCS, small residues at the bottom of the active site (V222, N226, A313, A316 and S320) form a similar tunnel for two non-reacting isoprene units as observed for TvTS. **c**, EI-MS ion chromatograms of extracts from (i) *S. cerevisiae* RC181 expressing the gene for CgCS and (ii) *S. cerevisiae* YZL141 (negative control). **d**, Proposed mechanism for the cyclization of HexPP to **3** through *syn* addition of C1 and water to the C14=C15 double bond.

together, the face selectivity at C23 and the stereospecificity of the deprotonation at C1 suggest an absolute configuration of (1*R*,22*R*)-**2**, which was confirmed by stereoselective deuteration through incubation of MpMS with (*R*)-(1-$^{13}$C,1-$^{2}$H)IPP and (*S*)-(1-$^{13}$C,1-$^{2}$H)IPP (Supplementary Fig. 35).

The structural basis of the cyclization mechanism was investigated by determination of the crystal structure of apo TvTS-TC by soaking with the non-reactive substrate surrogate 2,3-dihydro-HexPP (Extended Data Table 1 and Supplementary Scheme 2). Similarly to PaFS-TC[35], TvTS-TC adopts the characteristic class I TS fold[12,36] (Extended Data Fig. 1a, b), but TvTS-TC possesses a larger active site to accommodate HexPP. Although 2,3-dihydro-HexPP was not clearly observed and we cannot completely exclude the possibility that the observed electron density originated from polyethylene glycol (PEG) used in the crystallization buffer (Supplementary Fig. 36), the disordered regions in the apo structure, especially the DDXXD motif and the active site loop D173–D182, appeared clearly structured following soaking with the substrate surrogate. This observation suggests that a major conformational change occurs in TvTS following substrate binding that facilitates active site closure (Extended Data Fig. 1a, c). The docking model of 2,3-dihydro-HexPP based on the observed electron density suggested two possible conformers (Fig. 3a and Supplementary Fig. 36e), one in which HexPP is stretched out across the active site (yellow) and the other in which HexPP is prefolded (purple) for C1-III-IV cyclization with C1−C11 and C10−C14 distances of 5.4 and 3.2 Å, respectively.

The TvTS active site contains the conserved DDXXD and NSE motifs and other residues that interact with the substrate's diphosphate (Fig. 3a, b). Aromatic residues in TvTS-TC (F65, F89, F187, W188 and W310) are observed in positions analogous to those in PaFS (F65, F89, F196, W197 and W319), while the PaFS residues at the bottom of the active site (M221, W225, M309, S312 and N316) are substituted with smaller residues in TvTS (A212, Y216, G300, G303 and S307; Fig. 3a, b). These differences create a larger pocket in TvTS-TC that is composed of a tunnel to accommodate two additional (non-reacting) isoprene units of HexPP, sticking out from a ball-shaped cavity in which the four reacting isoprene units are located. This ball-shaped part is similar in shape and size to the PaFS active site that houses GGPP.

Structure-based site-directed mutagenesis experiments (Extended Data Fig. 2) included several small-to-large substitutions in TvTS near C15 to C20 of 2,3-dihydro-HexPP (G184F, A219L, G220L and S307F). The enzyme variants showed strongly reduced production of **1**, suggesting that the available space in this region is important for correct substrate folding. In addition, the exchange of conserved aromatic residues for alanine or leucine (F65A/L, F89A/L, F187A/L, W188A/L and W310A) abolished or decreased enzyme activity, indicating that these TvTS residues have an important role in shaping the active site cavity and/or stabilizing intermediates through cation−π interactions (Fig. 2e)[6]. The small residues that widen the binding pocket in comparison with PaFS were substituted with the corresponding residues in PaFS-TC. While the G303S and S307N substitutions had little effect, the A212M and Y216W variants retained only 15% and 28% of wild-type activity for the generation of **1** and production with the G300M variant was completely disrupted, in support of the hypothesis that the larger steric bulk introduced in these enzyme variants does not allow uptake of HexPP.

Further insights into triterpene biosynthesis by PT−TC chimeric enzymes were obtained by cryo-electron microscopy (cryo-EM) of MpMS. The data obtained for full-length MpMS showed that the enzyme exclusively forms a hexamer, whereas PaFS forms octamers and hexamers with a ratio of 9:1 (Fig. 3d, Supplementary Figs. 37 and 38 and Extended Data Table 2)[37]. A three-dimensional (3D) reconstruction of unliganded MpMS could be established for only the PT domain, as in the cryo-EM analysis of PaFS[37]. The overall structure of the MpMS-PT monomer is highly similar to the PaFS-PT cryo-EM and crystal structures, and the hexamer of MpMS-PT also superimposes well with the PaFS hexamer (root mean square deviation of 1.4−1.6 Å for 274 Cα atoms). Moreover, the active site of MpMS-PT is of suitable size for its product HexPP (Extended Data Fig. 3).

Interactions between the PT and TC domains in MpMS were further investigated by cross-linking the domains with glutaraldehyde[38]. The obtained structural data for cross-linked MpMS were substantially different from those for non-cross-linked MpMS (Supplementary Figs. 39 and 40 and Extended Data Table 2). Additional electron densities were observed in cross-linked MpMS at each vertex of the PT hexamer with a different occupancy, suggesting that the orientation of the TC domain

is flexible although its position was fixed by the cross-linking. However, because the local resolution was low and fragmented, a model for only one helix of the TC domain in addition to the PT domain could be built (Fig. 3e). Fitting of a TC domain homology model into the density map (Fig. 3f) suggested that the α5 helix region (residues 146–163) participates through cross-linking of K150 to K442 of the PT domain (the distance between these residues is only 3.0 Å). Moreover, TC residues T147, K150, N151 and K155 on helix α5 form hydrogen bonds with PT domain residues on helices α2 (419–429) and α14 (686–695; Extended Data Fig. 4).

Bifunctional TSs with an αα domain architecture may have a catalytic advantage via proximity channelling if the active sites of the interacting domains are properly faced towards each other[39–41]. In the PaFS octamer, this is ideally realized, as the TC domains of PaFS are oriented towards the central pores of the PT domains, which facilitates product channelling from the PT domain to the TC domain[37]. Additionally, in the hexameric structure of MpMS, each TC domain is located close to a PT domain, with their active sites facing each other (Extended Data Fig. 5), which allows for direct transfer of HexPP from the PT domain to an adjacent TC domain.

The MpMS-TC model[42] does not show the same tunnel for uptake of two non-reacting isoprene units as was observed for TvTS, as the TvTS positions of A212, G300 and G303 are occupied by bulky residues in MpMS (L238, L318 and M321). By contrast, the ball-shaped part of the active site of MpMS is wider because residues F187 and W188 of TvTS are substituted by smaller ones in MpMS (V206 and A207; Fig. 3a, c). These observations may explain why TvTS-TC with its initial C1-III-IV cyclization behaves like a diterpene synthase (DTS), acting on only the first four isoprene units of HexPP, while the two isoprene units at the end of the HexPP chain stick out into a tunnel and do not participate in terpene cyclization. By contrast, MpMS lacks this tunnel but has an overall larger reaction chamber for full uptake of HexPP, thus allowing its unusual C1-VI cyclization. In agreement with this hypothesis, MpMS V206F and A207W variants showed nearly or completely abolished production of **2** (Extended Data Fig. 6).

The high similarity between the AlphaFold2-predicted model and the crystal structure of TvTS-TC offers a basis for the discovery of additional chimeric fungal TrTSs through structural prediction (Fig. 4a and Supplementary Fig. 41), which is not possible from amino acid sequence analyses. Such structure predictions with docking of HexPP were performed for ten chimeric TSs with low sequence similarity to previously characterized enzymes. Six of these enzymes (PTTC027, PTTC044, PTTC060, PTTC074, PTTC114 and Cgl13855) contained a pocket that may be sufficiently large for HexPP binding (Supplementary Figs. 42 and 43 and Supplementary Table 7). Functional characterization through expression in yeast indeed resulted in colleterpenol (**3**) for Cgl13855 (NMDCN0000R73) from the endophyte *C. gloeosporioides* ES026 (ref. [43]) (Fig. 4c, Extended Data Fig. 7 and Supplementary Fig. 44), characterizing the TrTS as CgCS. HexPP docking identified an active site cavity for Cgl13855-TC similar to that of TvTS, explaining the C1-III-IV cyclization with two isoprene units sticking out into a tunnel (Fig. 4b).

The planar structure of **3** was identified by NMR (Supplementary Table 8 and Supplementary Figs. 45–59), showing structural similarity to the known 15-hydroxy-α-cericerene[44]. Solving the relative configuration of **3** through nuclear Overhauser enhanced spectroscopy (NOESY) was difficult because of its conformational flexibility. A comparison of calculated to measured NMR data favoured the structure (14*R*\*,15*S*\*)-**3**, which was further supported by a comparison of the ¹³C-NMR data to those reported for 15-hydroxy-α-cericerene[44] with the same relative configuration (Supplementary Table 8). The absolute configuration of (14*R*,15*S*)-**3** was then determined through comparison of measured and calculated electronic circular dichroism (ECD) curves (Supplementary Fig. 47). Notably, **3** formation requires *syn* addition of C1 and water to the C14=C15 alkene in HexPP (Fig. 4d). Additionally, PTTC074 produced a similar triterpene compound as CgCS (Extended Data Fig. 7 and Supplementary Fig. 44), which implies the existence of more TrTSs to be explored.

In summary, a novel family of chimeric class I TrTSs converting HexPP into triterpenes was identified for which two subclasses can be distinguished. Some TrTS-TCs provide one large cavity for HexPP uptake to form macrocycles. Other enzymes have a ball-shaped cavity of similar size to that in DTSs with an adjacent tunnel that accommodates two 'spectator' isoprene units, with the consequence that only the first four units participate in cyclization. This reflects the situation for the C1-III-IV and C1-IV-V cyclizing subclasses of sesterterpene synthases, but structural insights are lacking to understand these different modes. The findings described here show expanded product boundaries for class I TSs and enrich understanding of terpene biosynthesis in nature.

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

## Methods

### Strains and culture conditions

Cloning was performed in *Escherichia coli* DH10B using standard recombinant DNA techniques. *E. coli* BL21(DE3) (Invitrogen) was used for protein expression. Both *E. coli* strains were grown in LB (10.0 g l$^{-1}$ tryptone, 5.0 g l$^{-1}$ yeast extract, 5.0 g l$^{-1}$ NaCl, pH 7.0) at 37 °C. *S. cerevisiae* YZL141 (ref. [45]) was used for the heterologous expression of ZTR_06220 (KUL85185) from *T. verruculosus* TS63-9 (ref. [21]), MPH_02178 (EKG20455) from *M. phaseolina* MS6 (ref. [22]) and Cgl13855 (NMDCN0000R73) from *C. gloeosporioides* ES026 (ref. [43]) and grown in YPD medium (20 g l$^{-1}$ tryptone, 10 g l$^{-1}$ yeast extract, 20 g l$^{-1}$ glucose, pH 7.2) at 30 °C. *T. verruculosus* TS63-9 was selected for functional verification of TvTS in vivo and grown in PDB medium (20 g l$^{-1}$ potato extract, 20 g l$^{-1}$ glucose, pH 6.5).

### Phylogenetic analysis of TvTS and MpMS

To characterize the evolutionary relationships of TvTS, MpMS and CgCS, 56 characterized fungal chimeric class I TSs were selected and multiple-sequence alignment was performed using ClustalW (version 2.0.12). The Poisson correction model based on the maximum-likelihood method was used to infer the evolutionary history of these enzymes, and MEGA7 was used to conduct the evolutionary analysis[46]. Bootstrap values were obtained using 1,000 replications. The initial tree for the heuristic search was acquired automatically by applying the Neighbor-Join and BioNJ algorithms to a matrix of pairwise distances estimated using a JTT model and then selecting the topology with a superior log-likelihood value. After eliminating all positions that contained gaps and missing data, 186 positions were left in the final dataset. The plant-derived sesterterpene synthase AtTPS25 (ref. [47]) was selected as an outgroup.

### Functional characterization of TrTS candidates in vivo

Synthetic genes encoding TvTS (ZTR_06220), MpMS (MPH_02178), CgCS and the nine remaining candidates (Supplementary Table 7), codon optimized for expression in yeast, were obtained from GenScript. After digestion with HindIII and ScaI, the genes were cloned into HindIII- and ScaI-digested pYJ117 plasmid[20] to produce plasmids pXM139 (from *ZTR_06220*), pXM018 (from *MPH_02178*), pRC181 (from *Cgl13855*) and five additional plasmids. The plasmids were individually transformed into *S. cerevisiae* YZL141 to yield corresponding strains, which were grown in 5 ml YPD medium (2% glucose) at 30 °C overnight. From this starter culture, 5 ml was added to a 500-ml flask containing 200 ml YPD (2% glucose and 1% galactose), followed by growth at 30 °C with shaking at 220 r.p.m. for 3 d. Strains were collected, and the mycelium was collected and extracted with hexane/ethyl acetate (4:1). The organic layers were combined for gas chromatography mass spectrometry (GC–MS) analysis.

### Fermentation of engineered *S. cerevisiae* strains and isolation of 1 and 3

For isolation of **1** and **3**, *S. cerevisiae* XM139 and RC181 strains were scaled up in 2-litre shaker flasks containing 1 litre YPD (2% glucose and 1% galactose). Subsequently, the mycelium was collected and extracted with acetone three times. The acetone-extracted layer was distilled under reduced pressure to remove acetone and then partitioned between ethyl acetate and water to afford the ethyl acetate fraction (1.7 g for **1** and 1.56 g for **3**). For **1**, the residue was subjected to a silica gel column (80–100 mesh) and elution with petroleum ether/ethyl acetate (100:0, 99.8:0.2, 99:1, 98:2, 95:5, 0:100) to give fractions A–F. Following this, **1** was identified in fraction B by GC–MS detection and further purified by semi-preparative HPLC (Ultimate 3000 HPLC equipped with an XBridge Prep C18 column (Waters, 113; 10 × 250 mm, 5 μm)) to afford compound **1** (7.2 mg). For **3**, the crude extract was dissolved in methanol/DMSO (10:1, vol/vol) and subjected to semi-preparative HPLC (column: Agilent ZORBAX BS-C18, 5 μm, 9.6 × 250 mm internal

diameter; solvent: acetonitrile/H$_2$O, 99:1; flow: 3 ml min$^{-1}$; detector: 210, 230 nm) to yield compound **3** (30.5 mg, $t_R$ = 24.2 min).

### Plasmid construction for in vitro and labelling experiments

For expression of recombinant TvTS and MpMS in *E. coli*, codon-optimized synthetic genes were obtained from GenScript. The synthetic gene for MpMS was digested with NdeI and EcoRI and cloned into pET28a to produce pRC088. The sequences encoding the PT and TC domains of TvTS were amplified separately by PCR with Phusion DNA polymerase using primer pairs P1/P2 and P3/P4 (Supplementary Table 2). The amplified nucleotide sequence for the PT domain was digested with NdeI and EcoRI and cloned into pET28a to produce plasmid pRC009, and the amplified nucleotide sequence for the TC domain was digested with HindIII and XhoI and cloned into pET21a to generate plasmid pRC041. The sequence encoding the TC domain-inactivated MpMS D114A/N115A variant was amplified from pRC088 using primer pairs P5/P6 and P7/P8 (Supplementary Table 2) and then assembled by overlap extension PCR and cloned into NdeI- and EcoRI-digested pET28a to produce pRC088-D114A/N115A. Correct gene insertion was verified by sequencing, and plasmids were used to transform *E. coli* BL21(DE3) competent cells using a calcium-based protocol.

### Protein expression and purification for in vitro assays

For gene expression, a fresh LB culture of *E. coli* BL21(DE3) transformants (containing 100 mg l$^{-1}$ ampicillin for pRC041 (TvTS-TC) and 50 mg l$^{-1}$ kanamycin for pRC009 (TvTS-PT), pRC088 (MpMS) and pRC088-D114A/N115A) was inoculated from a glycerol stock and grown overnight. The precultures were used to inoculate the desired volume of LB (1 ml per litre) amended with the appropriate antibiotic. Cultures were grown at 37 °C with shaking until an OD$_{600}$ of 0.6–0.8 was reached. The cultures were cooled to 16 °C, and isopropyl β-D-1-thiogalactopyranoside (0.1 mM) was added to induce expression. Proteins were expressed overnight (~20 h) at 16 °C with shaking. Cells were collected by centrifugation (8,000g, 5 min). The supernatant was discarded, and the cell pellet was resuspended in buffer A (50 mM Tris-HCl, 300 mM NaCl, 4 mM β-mercaptoethanol, pH 7.6; 10 ml per litre of culture). Cells were lysed by sonication on ice (5 × 30 s). The cellular debris was removed by centrifugation (14,710g, 2 × 7 min), and the supernatant was subjected to Ni-NTA affinity chromatography (Protino Ni-NTA, Macherey-Nagel) through a syringe filter. The resin was washed with buffer A (20 ml per litre of culture), followed by elution of the His$_6$-tagged proteins using elution buffer (buffer A + 300 mM imidazole; 10 ml per litre of culture). The proteins were concentrated using centrifugal filters (10-kDa cut-off; 5,000g, 4 °C; Amicon Ultra-15 (Millipore) or Vivaspin 20 (Sartorius)) and diluted with incubation buffer (50 mM Na$_2$HPO$_4$, 10% glycerol, 2 mM MgCl$_2$). Enzyme concentrations were determined by Bradford assay and adjusted to 20 μM. Incubation experiments were carried out at 30 °C overnight using combinations of enzymes and substrates as listed in Supplementary Table 5. For experiments with unlabelled substrates, hexane (650 μl) was used for extraction, whereas for experiments with labelled substrates extraction was performed using C$_6$D$_6$ (650 μl). Samples were directly analysed by GC–MS and/or NMR spectroscopy.

### In vitro enzyme assays for TvTS-PT and MpMS D114A/N115A and detection of HexPP

Reactions were carried out using substrate (IPP and DMAPP, 100 μM each), 2 mM Mg$^{2+}$, 10% glycerol and 10 μM enzyme (TvTS-PT or MpMS-PT D114A/N115A) in Tris-HCl buffer (200 μl; 50 mM, pH 7.6) at 30 °C overnight. The resulting HexPP was extracted with acetonitrile and analysed by liquid chromatography mass spectrometry (LC–MS). For high-resolution MS analysis of HexPP, an LTQ Orbitrap Elite instrument coupled to a Thermo Scientific Ultimate 3000 RSLC HPLC system and an ACE UltraCore 2.5 SuperC18 (2.1 × 100 mm) column was used for compound separation at 35 °C. The mobile phase (pH 9.5) containing

5 mM ammonium bicarbonate in water as solvent A and acetonitrile as solvent B was set to a flow rate of 0.2 ml min$^{-1}$. The gradient programme was as follows: 98–10% solvent A (0–10 min), 10–0% solvent A (10–15 min), 0% solvent A (15–17 min), 0–98% solvent A (17–18 min) and 98% solvent A (18–22 min). EI was used in negative mode for detection. The ion source parameters were as follows: sheath gas, 40 arb; auxiliary gas, 5 arb; spray voltage, 3.1 kV; capillary temperature, 270 °C; S-lens RF level, 65 kV; auxiliary gas heater temperature, 250 °C. Full-scan MS mode with a resolution of 60,000 was used for qualitative analysis.

## Site-directed mutagenesis and in vitro analysis of enzyme variants

Site-directed mutagenesis for the construction of MpMS and TvTS enzyme variants was performed with the PCR-based QuikChange Site-Directed Mutagenesis kit (Stratagene) according to the manufacturer's protocol, using Phusion DNA polymerase and the mutational primers listed in Supplementary Table 2. Plasmids pRC088 (containing the full-length gene for MpMS) and pMBP139 (containing the full-length gene for TvTS) were used as template. All mutants were verified by gene sequencing. Reactions for TvTS, MpMS and the variants were carried out using substrate (DMAPP and IPP, 100 µM and 500 µM, respectively), 2 mM Mg$^{2+}$, 5% glycerol and 10 µM enzyme in Tris-HCl buffer (200 µl; 50 mM, pH 7.5) at 30 °C overnight. The resulting product was extracted with ethyl acetate and analysed by GC–MS. All experiments were performed in three biological replicates.

## Isolation of macrophomene (2) from incubation with MpMS E104Y

MpMS E104Y was expressed in *E. coli* and purified by Ni-NTA affinity chromatography. The pooled enzyme fractions were concentrated, further purified by size-exclusion chromatography and concentrated again. Incubation was performed using GPP (30 mg, 0.08 mM) and IPP (100 mg, 0.34 mM) dissolved in NH$_4$HCO$_3$ buffer (25 mM, 20 ml), which was added to the enzyme solution in incubation buffer (100 ml) over a period of 1 h using a syringe pump. The mixture was incubated overnight and extracted with *n*-hexane (150 ml) three times. The organic layers were dried with MgSO$_4$ and concentrated under reduced pressure. The residue was purified by column chromatography on SiO$_2$ (pentane) to yield macrophomene as a colourless oil.

## Construction of expression plasmids for protein crystallization

To increase the solubility of protein in *E. coli*, the coding sequence for TvTS was amplified using primer pair P21/P22 from the codon-optimized synthetic gene and ligated into NdeI- and HindIII-digested pET28-MBP-TEV, using the In-Fusion HD Cloning kit (TaKaRa), yielding plasmid pMBP139. For expression of protein for crystallization, the coding sequence for TvTS-TC was amplified from pRC041 using primer pair P23/P24. The amplified sequence was cloned into pET-SUMO, which was itself amplified with primer pair P25/P26, using the In-Fusion HD Cloning kit, resulting in plasmid pSUMO041.

## Protein expression and purification for crystallization

Protein expression was performed with *E. coli* BL21(DE3) harbouring pSUMO041 using the same protocol as described above. For crystallization, after Ni-NTA purification, SUMO–TvTS-TC was dialysed against 2 × 1 litre of Tris-HCl buffer (pH 8.0) containing 5% (vol/vol) glycerol and 300 mM NaCl. After dialysis, SUMO–TvTS-TC was treated with SUMO protease Ulp1403-621 (ref. [48]; prepared as previously described, 0.87 µM) in the presence of dithiothreitol (DTT, 1 mM) at 4 °C overnight. The protein solution was loaded onto a column filled with Ni-NTA resin. The His$_6$–SUMO fragment and protease were then captured by the Ni-NTA resin, leaving TvTS-TC in the flow-through, and the remaining protein on the column was eluted with Tris-HCl buffer (pH 8.0) containing 5% (vol/vol) glycerol, 10 mM imidazole and 300 mM NaCl. The collected protein solution was incubated with 10 mM EDTA (pH 8.0)

at 4 °C for 1 h. To obtain a protein preparation of high purity, further purification was performed using anion-exchange chromatography on a Resource Q column (Cytiva) by linearly increasing the salt concentration from 0 M NaCl to 1 M NaCl in buffer (50 mM Tris-HCl (pH 8.0), 1 mM DTT, 5% glycerol) over 20 column volumes. The desired protein was collected and then purified to homogeneity by size-exclusion chromatography on a HiLoad 16/600 Superdex 200 pg column (Cytiva) and eluted with a solution containing 20 mM Tris-HCl (pH 8.0), 100 mM NaCl, 1 mM DTT and 5% glycerol. The resulting eluate was concentrated to 10 mg ml$^{-1}$, using an Amicon Ultra-4 filter (molecular weight cut-off of 30 kDa) at 4 °C. The purity of the proteins was monitored by SDS–PAGE, and protein concentrations were determined with a SimpliNano microvolume spectrophotometer.

## Crystallization and structure determination

Crystals of TvTS-TC were obtained after 1 d at 10 °C by using the sitting-drop vapor-diffusion method. Before crystallization, 125 µM of protein was incubated with 2 mM MgCl$_2$ on ice for 30 min, and 0.5 µl of protein solution was then mixed with 0.5 µl of reservoir solution containing 0.1 M Tris-HCl (pH 8.5), 0.1 M MgCl$_2$, 30% PEG 4000 and 0.2 M NDSB-211. Crystals of TvTS-TC in complex with 2,3-dihydro-HexPP were obtained by incubation of TvTS-TC crystals with 10 mM 2,3-dihydro-HexPP in the crystallization drop at 10 °C for 14 h. The crystals were transferred to cryoprotectant solution (reservoir solution with 25% (vol/vol) glycerol) and then flash cooled at −173 °C in a nitrogen gas stream. The X-ray diffraction datasets were collected at X06SA (Paul Scherrer Institut, Villigen, Switzerland) for the apo TvTS-TC structure and at BL-1A (Photon Factory, Tsukuba, Japan) for the structure of TvTS-TC in complex with 2,3-dihydro-HexPP, using a beam wavelength of 1.0 and 1.1 Å, respectively. The diffraction datasets for TvTS-TC were processed and scaled using the XDS package[49] and Aimless[50]. The initial phase of the TvTS-TC structure was determined by molecular replacement, using PaFS (PDB, 5ER8) as the search model. Molecular replacement was performed with Phaser in PHENIX (version 1.19.2-4158-000)[51,52]. The initial phase was further calculated with AutoBuild in PHENIX[52]. The TvTS-TC structures were modified manually with Coot[53] and refined with PHENIX.refine[54]. The cif parameters of the ligands for the energy minimization calculations were obtained by using the PRODRG server[55]. After soaking with 2,3-dihydro-HexPP, strong additional electron densities were observed close to the DDXXD motif and in the active site cavity (Supplementary Fig. 36a, b). Modelling of 2,3-dihydro-HexPP to the observed density was partly satisfactory, but some of the methyl groups along the isoprenoid chain stuck out (Supplementary Fig. 36c). For an alternative explanation, PEG used in the crystallization buffer was modelled to the density, but in this case large unassigned densities close to the DDXXD motif remained in the refined structure (Supplementary Fig. 36d). Thus, it is possible that the observed density originated from both 2,3-dihydro-HexPP and PEG with low occupancies. The final crystal data and intensity statistics are summarized in Extended Data Table 1. The Ramachandran statistics were as follows: 97.6% favoured and 2.4% allowed for apo TvTS-TC, 98.9% favoured and 1.1% allowed for TvTS-TC soaked with 2,3-dihydro-HexPP. Although the ligand was not assigned, the conformations of PEG and 2,3-dihydro-HexPP in the active site should be similarly defined by active site residues. Therefore, a docking model of 2,3-dihydro-HexPP based on the observed density was developed. All crystallographic figures were prepared with PyMOL (DeLano Scientific; http://www.pymol.org).

## Purification of MpMS and cross-linking of the PT and TC domains

Protein expression was performed with *E. coli* BL21(DE3) harbouring pRC088 using the same protocol as described above. After Ni-NTA purification, MpMS protein solution was further purified using anion-exchange chromatography on a Resource Q column (Cytiva)

by linearly increasing the salt concentration from 20 mM NaCl to 1 M NaCl in buffer (50 mM HEPES (pH 7.5) and 1 mM DTT) over 20 column volumes. The desired protein was collected and then purified by size-exclusion chromatography on a Superose 6 10/300 column (Cytiva) using buffers containing 20 mM HEPES (pH 7.5), 150 mM NaCl and 1 mM DTT. The resulting eluate was concentrated to 50 μM, using an Amicon Ultra-4 filter (molecular weight cut-off of 100 kDa) at 4 °C. The purity of the proteins was monitored by SDS–PAGE. To obtain the structure of full-length MpMS, cross-linking of the PT and TC domains via glutaraldehyde (25% in water; Nacalai Tesque) was performed. First, MpMS protein was purified by Ni-NTA and anion-exchange chromatography, as described above, to obtain pure protein. Subsequently, cross-linking was performed on ice by incubating 1 mg ml$^{-1}$ MpMS protein with 0.06% glutaraldehyde for 10 min at a 100-μl scale. Reactions were quenched by adding 10 μl of 1.0 M Tris-HCl (pH 8.0). Reaction mixtures were pooled and further purified by size-exclusion chromatography on a Superose 6 10/300 column (Cytiva) to exclude aggregations. The resulting eluate was concentrated to 50 μM, using an Amicon Ultra-4 filter (molecular weight cut-off of 100 kDa) at 4 °C.

## Cryo-EM sample preparation and data acquisition
For cryo-grid preparation, 3 μl of sample was applied to a holey carbon grid (Quantifoil, Cu, R1.2/1.3, 300 mesh). The grid was rendered hydrophilic by a 30-s glow discharge in air (11-mA current) with PIB-10 (Vacuum Device). The grid was blotted for 20 s (blot force of 0) at 18 °C and 100% humidity and then flash frozen in liquid ethane using a Vitrobot Mark IV (Thermo Fisher Scientific). For the MpMS-PT and MpMS cross-linking datasets, 1,888 and 1,529 movies were acquired, respectively, on a Talos Arctica (FEI) microscope operating at 200 kV in nanoprobe mode using EPU software for automated data collection. The movies were collected on a 4,000 × 4,000 grid using a Falcon 3EC direct electron detector (electron counting mode) at a nominal magnification of 120,000 (0.88 Å per pixel). Fifty movie fractions were recorded at an exposure of 1.00 electrons per Å$^2$ per fraction, corresponding to a total exposure of 50 electrons per Å$^2$. The defocus steps used were −1.0, −1.5, −2.0 and −2.5 μm. The movie fractions were aligned, dose weighted and averaged using RELION's own implementation on 5 × 5 tiled fractions with a $B$-factor of 300. The non-weighted movie sums were used for contrast transfer function (CTF) estimation with Gctf [56]. The dose-weighted sums were used for all subsequent steps of image processing. The subsequent processes of particle picking, two-dimensional classification, ab initio reconstruction, 3D classification, 3D refinement, CTF refinement and Bayesian polishing were performed using RELION-3.18 (ref. [57]). For details of the cryo-EM data processing, see Supplementary Figs. 37–40.

## AlphaFold2 prediction and docking analysis
UCSF Chimera[58] (version 1.12) and AutoDock Vina[59] (version 1.1.2) were used to perform receptor and ligand preparation and molecular docking analysis. Before the docking procedure, the receptor structures predicted by AlphaFold2 and ligands were processed as follows. Metal ions were added to the TC domain binding site of receptors, referring to the homologous protein in PDB for the coordinates of the three Mg$^{2+}$ ions. With the exception of Cgl13855 and FgMS, for which 5IMP and 5ER8 were used as the reference for metal ions, respectively, 6VYD was used as the reference for the other PTTCs. The receptor structures containing metal ions were then processed using the Chimera tool Dock Prep, in which hydrogen atoms and charges were added and other parameters were set as default. The energy-minimization molecule models included in Chimera were used to minimize the energy for the structures from the previous step. The ligand structures were drawn, charge was added and the structures were transformed to 3D conformations. The binding site of the TC domain was determined by referring to the crystal structures for homologous proteins. The grid box in the docking procedure was defined to include the metal ions,

and corresponding residues appeared in the binding site of crystal structures for homologous proteins. Receptor and ligand options in AutoDock Vina were set as default. The number of binding modes, exhaustiveness of search and maximum energy difference (kcal mol$^{-1}$) parameters were set as 9, 8 and 3, respectively.

## Reporting summary
Further information on research design is available in the Nature Research Reporting Summary linked to this paper.

## Data availability
The authors declare that the main data supporting the findings of this study are available within the article and its Supplementary Information. Original data can be obtained from the corresponding authors on reasonable request. The coordinates and structure factor amplitudes for the apo structure of TvTS-TC and the structure of TvTS-TC after soaking with 2,3-dihydro-HexPP have been deposited to PDB under accession codes 7VTA and 7VTB, respectively. The cryo-EM maps and atomic coordinates for MpMS-PT and cross-linked MpMS have been deposited in the Electron Microscopy Data Bank (EMDB; https://www.ebi.ac.uk/pdbe/emdb/) and PDB with accession codes EMD-32531 and EMD-32532 and 7WIJ, respectively. The accession numbers (NMDC-N0000RG9, NMDCN0000RGA, NMDCN0000RGB, NMDCN0000R73, KFX89132, KAH9237577, KIK55704, KAF2708718, CRG86078, QIH97829) corresponding to the PTTC candidates for AlphaFold2 prediction have been deposited in the National Microbiology Data Center (https://nmdc.cn/en) and are listed in the Supplementary Information.

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

**Acknowledgements** This work was funded by the National Key R&D Program of China (2018YFA0900400 and 2021YFC2102600), the National Natural Science Foundation of China (31670090 and 31800032) and the German Research Foundation (DFG; DI1536/7-2). The numerical calculations in this paper were carried out on the supercomputing system in the Supercomputing Center of Wuhan University. The synchrotron radiation experiments were performed at BL-1A of the Photon Factory (proposal number 2019G038) and X06SA of the SLS (proposal numbers 20191094 and 20191134). This work was also supported in part by the Platform Project for Supporting Drug Discovery and Life Science Research (Basis of Supporting Innovative Drug Discovery and Life Science Research) from AMED (JP21am0101071 (support number 1553)), Grants-in-Aid for Scientific Research from the Ministry of Education, Culture, Sports, Science and Technology, Japan (JSPS KAKENHI grants JP16H06443, JP19K15703, JP20H00490, JP20KK0173, JP21K18246), the New Energy and Industrial Technology Development Organization (NEDO, grant JPNP20011), the PRESTO program from Japan

Science and Technology Agency (JPMJPR2ODA) and AMED (grant JP21ak0101164). H.T. is a recipient of the JSPS Postdoctoral Fellowship for Foreign Researchers (P18404).

**Author contributions** G.B. initiated the project and found the TrTS activity of TvTS and MpMS. R.C. and M.L. performed in vitro and in vivo analysis of TvTS and MpMS. S.C. verified the function of CgCS. X.M. constructed the plasmids and carried out fermentation. L. Lu and B.H. performed AlphaFold2 analysis and function prediction. Z.D. and X.W. supervised biological experiments. H.T. performed in vitro analysis and mutagenesis experiments. H.T. and T. Mori performed crystallization experiments. H.T., T. Mori, N.A., M.K., T. Moriya and T.S. performed the cryo-EM analysis. A.H. synthesized HexPP and 2,3-dihydro-HexPP. A.H. and L. Lauterbach performed in vitro characterization of and isotopic labelling experiments with TvTS-PT, TvTS-TC and MpMS. H.T., T. Mori, I.A., L. Lauterbach, A.H., J.S.D., G.B., R.C. and T.L. analysed the data and wrote the manuscript. I.A., J.S.D. and T.L. designed and supervised the study.

**Competing interests** T.L. has filed a patent application relating to the function of TvTS, MpMS and CgCS and the resulting structures talaropentaene and colleterpenol. The other authors declare no competing interests.

**Additional information**
**Correspondence and requests for materials** should be addressed to Ikuro Abe, Jeroen S. Dickschat or Tiangang Liu.

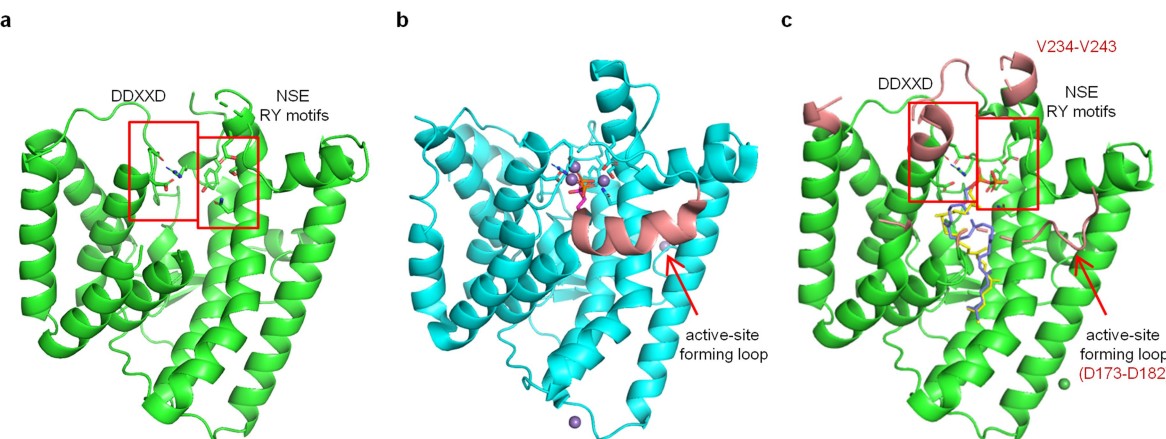

**Extended Data Fig. 1 | The structure of cyclase domain of TvTS. a**, the apo-form of TvTS-TC (green). The DDXXD, NSE, and RY motifs (red box) are conserved, and the active site forming regions, including aspartic-rich metal binding DDXXD motif, the region (V234-V243) after the NSE motif, and the A156-C183 region are disordered; **b**, PaFS-TC in complex with neridronate (cyan cartoon with magenta sticks, PDB: 5ER8[35], in comparison to TvTS-TS: rmsd values of 1.6 Å for Cα-atoms, 47% amino acid sequence identity); **c**, partially closed conformation of TvTS-TC. The disordered regions in the apo structure, especially the DDXXD motif and the active site loop D173-D182 (shown in salmon), appear clearly structured after soaking with 2,3-dihydro-HexPP. The docking model of 2,3-dihydro-HexPP was constructed based on the observed density and two possible conformers are shown by yellow and purple sticks.

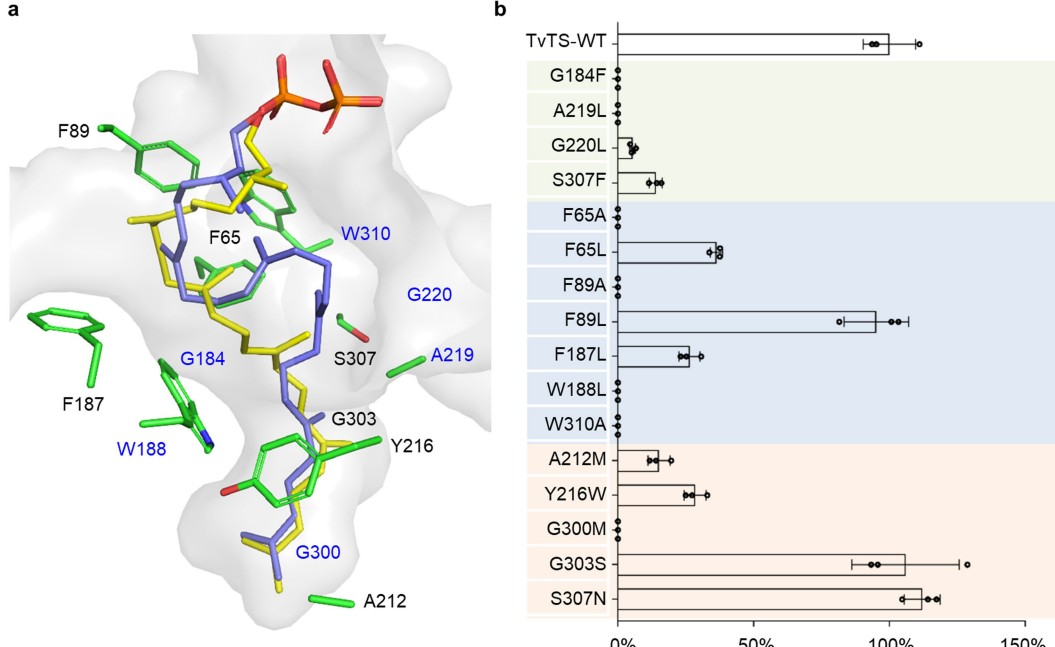

**a**

**b**

**Extended Data Fig. 2 | Structure based mutagenesis studies with TvTS.**
**a**, Active site cavity of TvTS-TC. The docking model of 2,3-dihydro-HexPP was constructed based on the observed density and two possible conformers were shown as yellow and purple sticks. **b**, In vitro activities of wild-type TvTS and its variants for production of **1**. Full-length TvTS and its variants were expressed with a maltose binding protein (MBP) fused at the N-terminus. Peak integrals for the ion chromatogram of $m/z$ 408 were used for quantification. Wild-type production was set to 100%, bars and error bars show mean and s.d. from three biological replicates, respectively.

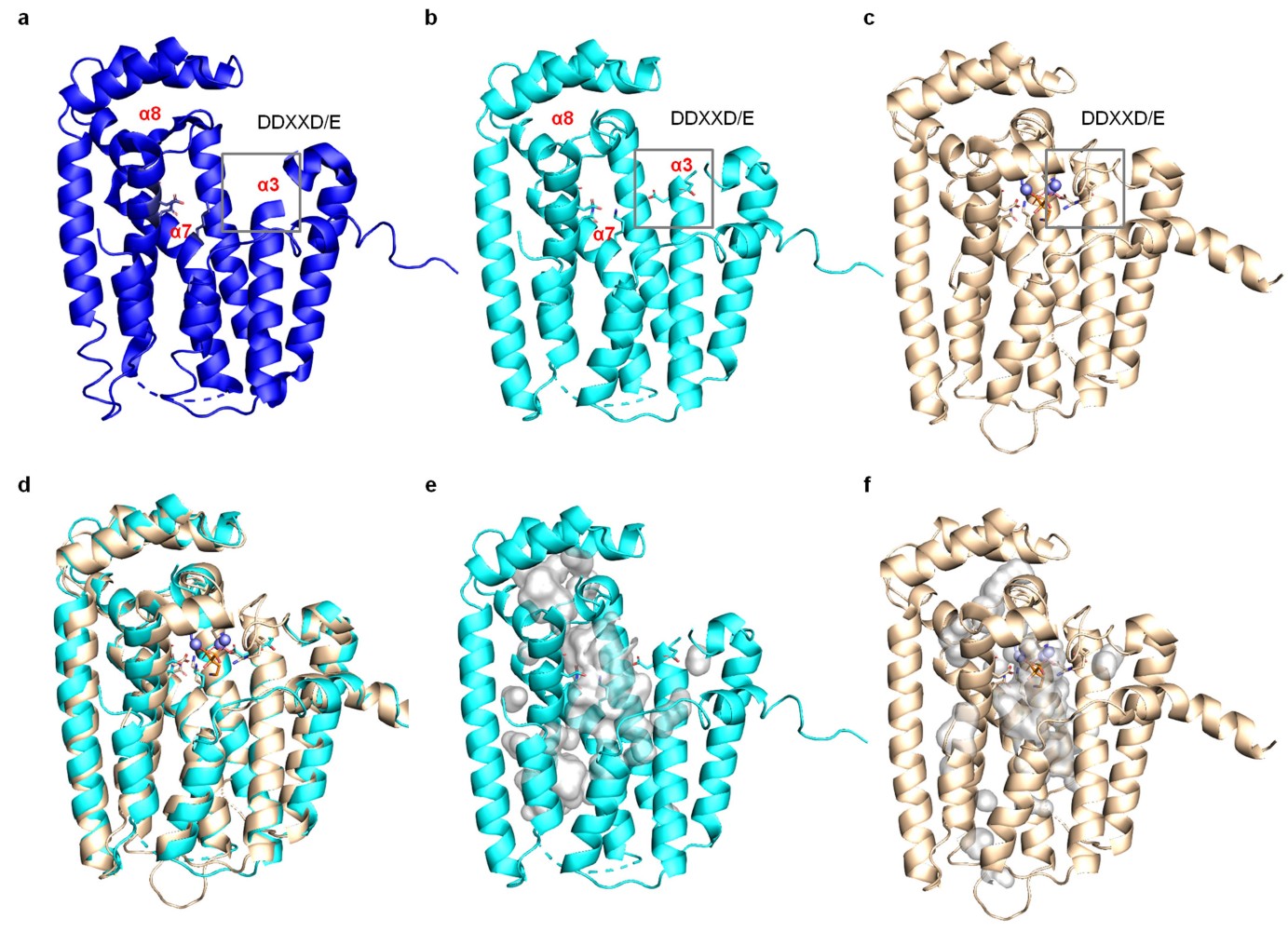

**Extended Data Fig. 3 | Active site of prenyltransferase domain of MpMS.** **a**, noncrosslinked MpMS-PT; **b**, crosslinked MpMS-PT; **c**, crystal structure of PaFS-PT complexed with cobalt ions and pamidronate (PDB ID: 5ERO); **d**, superimposed crosslinked MpMS-PT (cyan) and PaFS-PT (wheat) crystal structure; **e**, active site cleft of MpMS-PT; **f**, active site cavity of PaFS-PT.

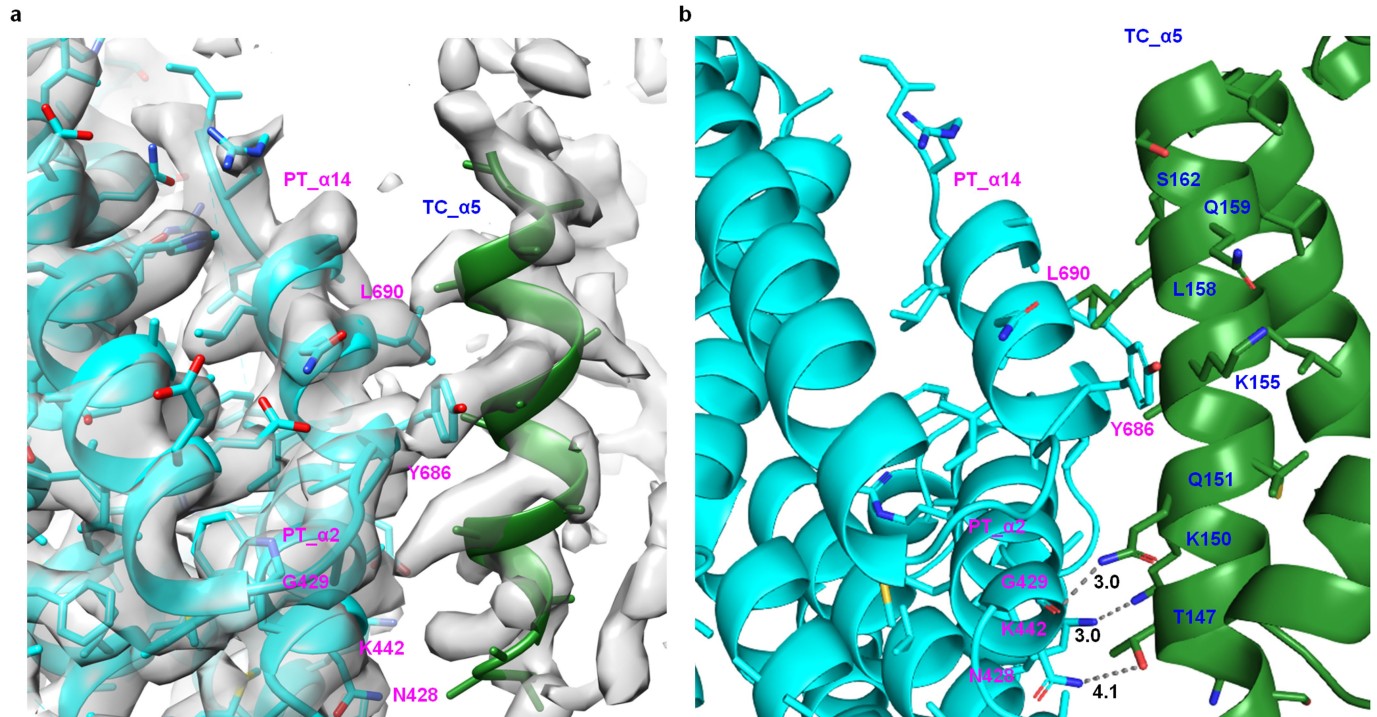

**Extended Data Fig. 4 | Close-up view of the interface between PT and TC domains. a**, The residues on α5 of TC domain (green) interact with the residues on α2 (419-429) and C-terminal α14 (686-695) of the PT domain (cyan);

**b**, close-up view of the interface between the PT domain and fitted model of TC domain. Distances indicated by grey dashed lines are given in Å.

**a**

**b**

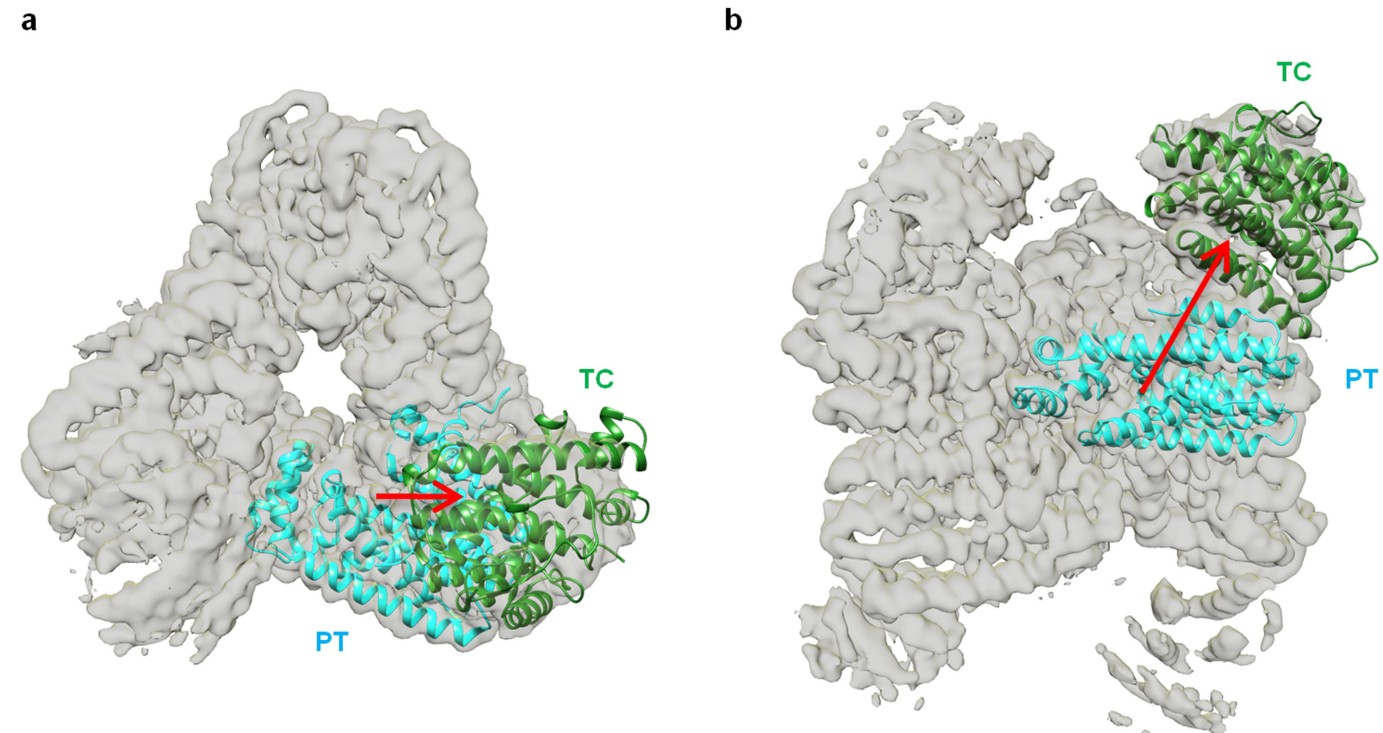

**Extended Data Fig. 5 | The active site of MpMS-PT (cyan) and TC (green) domains face to each other. a**, top view; **b**, side view.

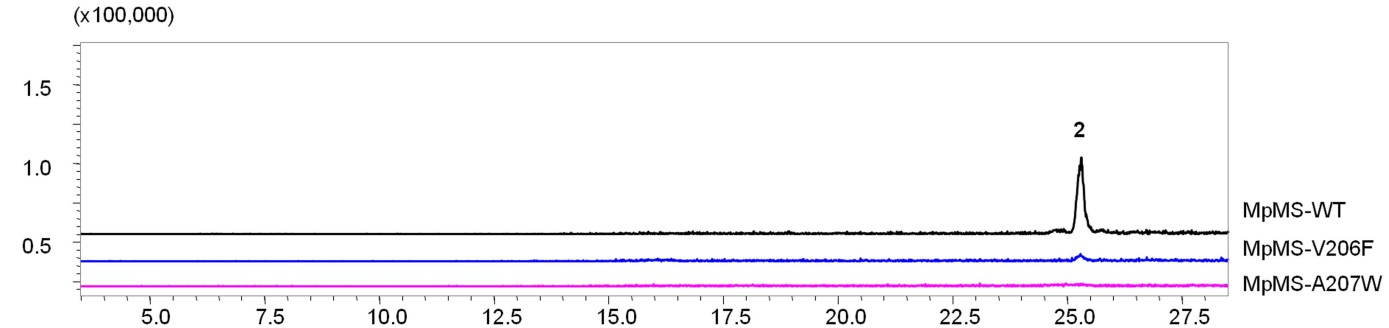

**Extended Data Fig. 6 | Mutagenesis study of MpMS.** Ion chromatograms ($m/z$ = 408) of wildtype MpMS (black) and its enzyme variants V206F (blue) and A207W (magenta).

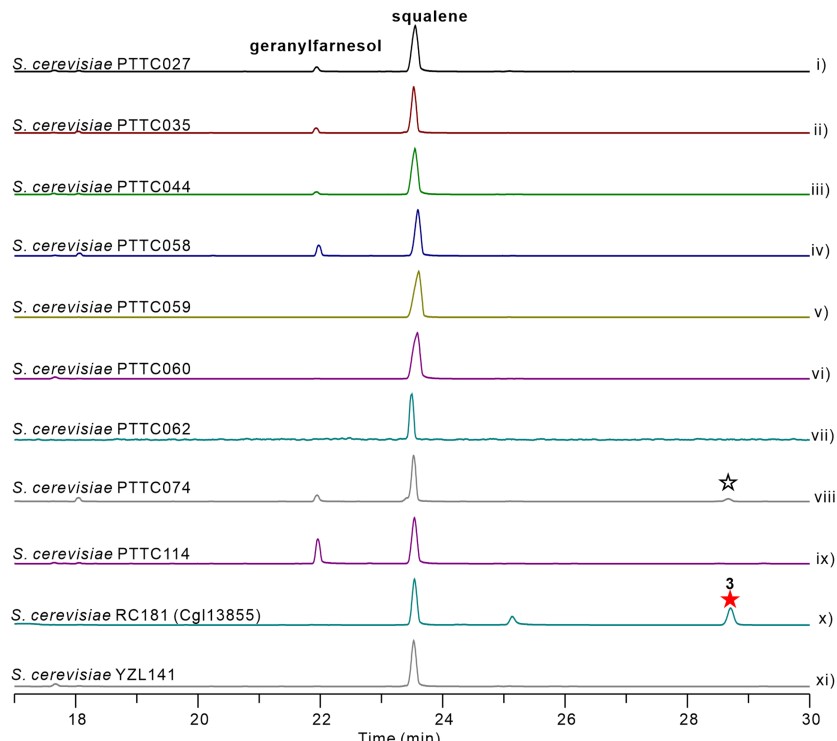

**Extended Data Fig. 7 | Product detection of triterpenes produced by chimeric class I TrTS in *S. cerevisiae*.** Engineered *S. cerevisiae* containing i) *PTTC027*, ii) *PTTC035*, iii) *PTTC044*, iv) *PTTC58*, v) *PTTC59*, vi) *PTTC60*, vii) *PTTC62*, viii) *PTTC74*, ix) *PTTC114*, x) *Cgl13855* overexpressed under the control of GAL10 promoter; xi) blank, engineered efficient terpene precursor providing chassis *S. cerevisiae* YZL141. Red asterisks indicated the presence of compound **3**, white asterisks indicate the similar peak produced by PTTC074.

**Extended Data Table 1 | Data collection and refinement statistics**

|  | TvTS apo | TvTS soaked with 2,3-dihydro-HexPP |
|---|---|---|
| **Data collection[a]** | | |
| Space group | C2 | C2 |
| Cell dimensions | | |
| $a$, $b$, $c$ (Å) | 76.5, 61.8, 81.7 | 75.8, 61.9, 81.3 |
| $\beta$ (°) | 116.3 | 116.3 |
| Resolution (Å) | 47.1–2.4 (2.49–2.40)[b] | 47.1–2.0 (2.05–2.00)[b] |
| $R_{merge}$ (%) | 9.7 (58.8)[b] | 5.0 (23.2)[b] |
| $I$ / $\sigma I$ | 13.3 (3.5)[b] | 14.4 4.2)[b] |
| Completeness (%) | 99.2 (99.6)[b] | 98.1 (97.4)[b] |
| Redundancy | 6.9 (6.8)[b] | 3.6 (3.7)[b] |
| | | |
| **Refinement** | | |
| Resolution (Å) | 47.1-2.4 | 47.2-2.0 |
| No. reflections | 13269 | 22445 |
| $R_{work}$ / $R_{free}$ | 23.1/27.0 | 18.1/22.4 |
| No. atoms | | |
| Protein | 1878 | 2255 |
| Ligand/ion | – | 1 |
| Water | 29 | 171 |
| $B$-factors | | |
| Protein | 50.6 | 26.4 |
| Ligand/ion | – | 26.1 |
| Water | 47.1 | 32.3 |
| R.m.s. deviations | | |
| Bond lengths (Å) | 0.007 | 0.007 |
| Bond angles (°) | 0.911 | 0.722 |
| | | |
| **Data access** | | |
| PDB | 7VTA | 7VTB |

[a] Data were collected from one crystal. [b] Values in parentheses are for highest-resolution shell.

**Extended Data Table 2 | Data collection and refinement statistics of cryo-EM data**

| | MpMS-PT domain (EMD-32531) (PDB: 7WIJ) | MpMS-crosslink (EMD-32532) |
|---|---|---|
| **Data collection and processing** | | |
| Microscope | Talos Arctica | Talos Arctica |
| Voltage (kV) | 200 | 200 |
| Detector | Falcon 3EC | Falcon 3EC |
| Magnification | 120,000 | 120,000 |
| Pixel size (Å) (calibrated) | 0.88 (-) | 0.88 (-) |
| Automation software | EPU | EPU |
| Total exposure (e–/Å$^2$) | 50 | 50 |
| Exposure rate (e–/Å$^2$ frame) | 1.00 | 1.00 |
| No. of frames | 50 | 50 |
| Defocus range (μm) | -1 to -2.5 | -1 to -2.5 |
| Symmetry imposed | D3 | C1 |
| No. of collected micrographs | 1,888 | 1,529 |
| No. of selected micrographs | 1,888 | 1,529 |
| No. of particles for 2D classification | 904,572 | 573,736 |
| No. of particles for 3D classification | 295,009 | 282,791 |
| No. of particles for 3D refinement | 132,926 | 113,860 |
| Map resolution (Å) | 3.17 | 4.00 |
| FSC threshold | 0.143 | 0.143 |
| Local resolution range (Å) | 3.00-3.78 | 3.50-9.64 |
| | | |
| **Refinement** | | |
| Initial model used (PDB code) | none | |
| Model composition | | |
| Non-hydrogen atoms | 12,390 | |
| Protein residues | 1602 | |
| Ligands | 0 | |
| *B* factors (min/max/mean, Å$^2$) | | |
| Protein | 2.6/69.0/21.9 | |
| Ligand | 0/0/0 | |
| R.m.s. deviations from ideal values | | |
| Bond lengths (Å) | 0.003 | |
| Bond angles (°) | 0.609 | |
| Validation | | |
| MolProbity score | 1.24 | |
| Clashscore | 4.72 | |
| Poor rotamers (%) | 0 | |
| Ramachandran plot | | |
| Favored (%) | 99.3 | |
| Allowed (%) | 0.7 | |
| Disallowed (%) | 0 | |

# Reporting Summary

## Statistics

For all statistical analyses, confirm that the following items are present in the figure legend, table legend, main text, or Methods section.

| n/a | Confirmed | |
|---|---|---|
| ☐ | ☒ | The exact sample size (*n*) for each experimental group/condition, given as a discrete number and unit of measurement |
| ☐ | ☒ | A statement on whether measurements were taken from distinct samples or whether the same sample was measured repeatedly |
| ☒ | ☐ | The statistical test(s) used AND whether they are one- or two-sided<br>*Only common tests should be described solely by name; describe more complex techniques in the Methods section.* |
| ☒ | ☐ | A description of all covariates tested |
| ☒ | ☐ | A description of any assumptions or corrections, such as tests of normality and adjustment for multiple comparisons |
| ☐ | ☒ | A full description of the statistical parameters including central tendency (e.g. means) or other basic estimates (e.g. regression coefficient) AND variation (e.g. standard deviation) or associated estimates of uncertainty (e.g. confidence intervals) |
| ☒ | ☐ | For null hypothesis testing, the test statistic (e.g. *F*, *t*, *r*) with confidence intervals, effect sizes, degrees of freedom and *P* value noted<br>*Give P values as exact values whenever suitable.* |
| ☒ | ☐ | For Bayesian analysis, information on the choice of priors and Markov chain Monte Carlo settings |
| ☒ | ☐ | For hierarchical and complex designs, identification of the appropriate level for tests and full reporting of outcomes |
| ☒ | ☐ | Estimates of effect sizes (e.g. Cohen's *d*, Pearson's *r*), indicating how they were calculated |

*Our web collection on statistics for biologists contains articles on many of the points above.*

## Software and code

Policy information about availability of computer code

| | |
|---|---|
| Data collection | CCREST 2.8, Gaussian 09, Thermo Xcalibur 2.2, Agilent MSD ChemStation D02.00.237, Shimadzu GCMS solution 4.41, EPU |
| Data analysis | MEGA7, MestReNova 5.3.1, PHENIX-ver 1.19.2-4158-000, ccp4-7.1, XDS ver Jan 31 2020, Coot 0.9, PyMOL ver 2.0.6, RELION-3.1, UCSF chimera 1.12.1 and 1.13.1, AutoDock Vina 1.1.2., Prism 9, Multiwfn 3.8 |

For manuscripts utilizing custom algorithms or software that are central to the research but not yet described in published literature, software must be made available to editors and reviewers. We strongly encourage code deposition in a community repository (e.g. GitHub). See the Nature Portfolio guidelines for submitting code & software for further information.

## Data

Policy information about availability of data

All manuscripts must include a data availability statement. This statement should provide the following information, where applicable:
- Accession codes, unique identifiers, or web links for publicly available datasets
- A description of any restrictions on data availability
- For clinical datasets or third party data, please ensure that the statement adheres to our policy

The authors declare that the main data supporting the findings of this study are available within the article and its Supplementary Information file. Original data can be obtained from the corresponding authors on reasonable request. The coordinates and the structure factor amplitudes for the apo structure of TvTS-TC and for the structure of TvTS-TC after soaking with 2,3-dihydro-HexPP were deposited under accession codes 7VTA and 7VTB, respectively. The cryo-EM maps and the atomic coordinates for MpMS-PT domain and MpMS-crosslink have been deposited in Electron Microscopy Data Bank (EMDB, https://www.ebi.ac.uk/pdbe/emdb/) and PDB with accession codes EMD-32531 and EMD-32532, and 7WIJ, respectively. The accession numbers (NMDCN0000RG9, NMDCN0000RGA, NMDCN0000RGB,

NMDCN0000R73, KFX89132, KAH9237577, KIK55704, KAF2708718, CRG86078, QIH97829) of PTTC candidates for AlphaFold2 prediction were deposited in the National Microbiology Data Center (https://nmdc.cn/en) and listed in supplementary information.

# Field-specific reporting

Please select the one below that is the best fit for your research. If you are not sure, read the appropriate sections before making your selection.

☒ Life sciences ☐ Behavioural & social sciences ☐ Ecological, evolutionary & environmental sciences

For a reference copy of the document with all sections, see nature.com/documents/nr-reporting-summary-flat.pdf

# Life sciences study design

All studies must disclose on these points even when the disclosure is negative.

| | |
|---|---|
| Sample size | Quantitative assays were performed in at least three independent biological replicates and mean and standard deviation values calculated. Such sample size was sufficient to determine the enzyme variant activity according to previous publications in a similar field. |
| Data exclusions | No data was excluded from the manuscript. |
| Replication | Reproducibility was verified by performing three or more independent biological replicates and noted. All attempts at replication were successful. |
| Randomization | No randomization was performed during this study as it was not applicable for our experiments. Detection of triperpene products were run consecutively on the GC-MS to minimize instrument drift within each sample. Single protein crystal structures were solved and randomization is not applicable to this section. |
| Blinding | No blinding was involved in this study as it does not involve animal or human subjects or group allocation. No class I triterpenes has been reported before, and data was collected and analyzed using software in an objective manner. No data was excluded from the analyses, so blinding is less relevant in this work. All data were analyzed and checked by multiple authors and reviewed by the corresponding author. |

# Reporting for specific materials, systems and methods

We require information from authors about some types of materials, experimental systems and methods used in many studies. Here, indicate whether each material, system or method listed is relevant to your study. If you are not sure if a list item applies to your research, read the appropriate section before selecting a response.

## Materials & experimental systems

| n/a | Involved in the study |
|---|---|
| ☒ ☐ | Antibodies |
| ☒ ☐ | Eukaryotic cell lines |
| ☒ ☐ | Palaeontology and archaeology |
| ☒ ☐ | Animals and other organisms |
| ☒ ☐ | Human research participants |
| ☒ ☐ | Clinical data |
| ☒ ☐ | Dual use research of concern |

## Methods

| n/a | Involved in the study |
|---|---|
| ☒ ☐ | ChIP-seq |
| ☒ ☐ | Flow cytometry |
| ☒ ☐ | MRI-based neuroimaging |

