## [Peer Review File · Nature]

Manuscript Title: Discovery of non-squalene triterpenes

Editorial Notes:

Redactions – unpublished data

Reviewer Comments & Author Rebuttals

Reviewer Reports on the Initial Version:

Referees' comments:

Referee #1 (Remarks to the Author):

The manuscript by Tao and colleagues outlines the discovery and characterization of bifunctional class I triterpene synthases, each of which contains a prenyltransferase and a cyclase. In contrast with currently known monofunctional class II triterpene cyclases that utilize squalene or squalene oxide as substrates, e.g., oxidosqualene cyclase in the cholesterol biosynthetic pathway [Thoma et al. (2004) Insight into steroid scaffold formation from the structure of human oxidosqualene cyclase. *Nature* 432, 118], the newly discovered triterpene cyclases use a metal-dependent mechanism to generate and then cyclize a 30-carbon substrate, hexaprenyl diphosphate. Metal-dependent chemistry here is identical to that employed by well-characterized class I monoterpene, sesquiterpene, and diterpene synthases (both prenyltransferases and cyclases). The discovery of class I triterpene cyclases is an important advance for the field of terpene biosynthesis.

Elegant chemical, computational, and enzymological studies using isotopically labeled substrates provide important mechanistic clues about the cyclization reactions catalyzed by the newly identified class I triterpene cyclases (TvTS, MpMS, and CgCS). These cyclization reactions are reminiscent of those encountered in the generation of cyclic monoterpenes, sesquiterpenes, and diterpenes. The biological relevance of each cyclization product is undefined: "the biological function and activity for these triterpenes and their potential derivatives is unknown and worth further attention" (lines 434-436).

Structural studies accompany the functional studies: the crystal structure of the cyclase domain of TvTS is reported, along with the cryo-EM structure of full-length MpMS. Both structures are well determined based on statistics recorded in Tables S7 and S8, respectively. The TvTS cyclase structure is homologous to structures of other class I terpene cyclases that utilize shorter substrates, but it has a deeper active site to accommodate the longer 30-carbon substrate. An electron density map of the complex with a substrate analogue is presented in Figure S36, but the map quality is poor and the fit of the model is not convincing. Moreover, it appears that not all of the olefin bonds are trans in the blue conformer here and also in Figure 2d (the fourth isoprenoid group appears to be cis; this should be a trans C=C bond).

The cryo-EM structure of full-length MpMS reveals density for a hexameric prenyltransferase, similar to that first observed for human geranylgeranyl diphosphate synthase [Kavanagh et al. (2006) The crystal structure of human geranylgeranyl pyrophosphate synthase reveals a novel hexameric

arrangement and inhibitory product binding. *J. Biol. Chem.* 281, 22004]. The cyclase domains of MpMS are disordered. As the authors note, this structure is similar to that of fusacoccadiene synthase (PaFS, ref. 33), which consists of a central prenyltransferase octamer or hexamer surrounded by randomly placed cyclase domains. Covalent crosslinking of MpMS with glutaraldehyde allowed for the identification of density corresponding to cyclase domains associated with the prenyltransferase hexamer (Figure 3). The cryo-EM structure of glutaraldehyde-crosslinked PaFS similarly revealed density for an associated cyclase domain which was confirmed using mass spectrometry (ref. 33). However, association sites are on the "top" and "bottom" of the MpMS prenyltransferase hexamer, whereas association sites are on the "side" of the PaFS octamer. Differences in prenyltransferase-cyclase association represent an important finding. The manner by which these two domains associate leads to alternative models for understanding substrate channeling in catalysis, as discussed on lines 332-342. The authors conclude that this study "broadens the boundaries of class I TSs functions and enriches our understanding towards terpene biosynthesis in nature" (lines 444-445).

In summary, the most important aspect of this work is the discovery of class I triterpene synthases that generate and cyclize the 30-carbon substrate hexaprenyl diphosphate. This broadens the family of class I terpene cyclases, both prenyltransferases and cyclases. Detailed studies of the newly identified class I triterpene cyclases reveal familiar structures and mechanistic features as previously encountered in other class I enzymes. The biological importance of the newly discovered triterpenes is yet to be determined.

Referee #2 (Remarks to the Author):

This is an excellent manuscript. The discovery of a family of alternative triterpene synthases that utilise a substrate other than squalene/2,3-oxidosqualene is a landmark advance. The body of work is substantial, impressive and very thorough. I have only a few questions/comments:

- 1) The authors chose ten chimeric TSs with low similarity to previous TSs and built their structures using AlphaFold2. Docking showed that five of these TSs have the potential to bind HexPP based on the finding that they had larger binding pockets. These five were tested experimentally for the production of triterpenes, of which one did indeed form a triterpene. It would be good to include experimental data for the other five that were not chosen based on their AlphaFold2 model, as it would act as a validation of the AlphaFold2 method.
- 2) Perhaps beyond the scope of this paper, but it would be interesting to investigate the expression of these genes at different developmental stages/under different environmental conditions (e.g. minimal nutrients v nutrient rich media) to try to gain some insight into when these novel triterpenes may be produced.
- 3) Is there any evidence that related enzymes might exist in plants?

Author Rebuttals to Initial Comments:

武汉大学

WUHAN UNIVERSITY

Discover
Excellence.

東京大学
THE UNIVERSITY OF TOKYO

We thank reviewers for their positive recommendation and valuable suggestions! We have carefully considered all the comments and provided our responses in a point-by-point format, and reduced the length of the article to meet the journal format requirements.

Referees' comments:

Referee #1 (Remarks to the Author):

The manuscript by Tao and colleagues outlines the discovery and characterization of bifunctional class I triterpene synthases, each of which contains a prenyltransferase and a cyclase. In contrast with currently known monofunctional class II triterpene cyclases that utilize squalene or squalene oxide as substrates, e.g., oxidosqualene cyclase in the cholesterol biosynthetic pathway [Thoma et al. (2004) *Insight into steroid scaffold formation from the structure of human oxidosqualene cyclase*. *Nature* 432, 118], the newly discovered triterpene cyclases use a metal-dependent mechanism to generate and then cyclize a 30-carbon substrate, hexaprenyl diphosphate. Metal-dependent chemistry here is identical to that employed by well-characterized class I monoterpene, sesquiterpene, and diterpene synthases (both prenyltransferases and cyclases). The discovery of class I triterpene cyclases is an important advance for the field of terpene biosynthesis.

Elegant chemical, computational, and enzymological studies using isotopically labeled substrates provide important mechanistic clues about the cyclization reactions catalyzed by the newly identified class I triterpene cyclases (TvTS, MpMS, and CgCS). These cyclization reactions are reminiscent of those encountered in the generation of cyclic monoterpenes, sesquiterpenes, and diterpenes. **The biological relevance of each cyclization product is undefined: "the biological function and activity for these triterpenes and their potential derivatives is unknown and worth further attention" (lines 434-436).**

Response: We thank the reviewer for giving positive comments on this work. At present, we don't know the biological function and activity for these triterpenes, we are very interested in and planning to characterize the biosynthetic pathway and the derived triterpenes, and investigate the biological function and activity in the near future.

Structural studies accompany the functional studies: the crystal structure of the cyclase domain of TvTS is reported, along with the cryo-EM structure of full-length MpMS. Both structures are well determined based on statistics recorded in Tables S7 and S8, respectively. The TvTS cyclase structure is homologous to structures of other class I terpene cyclases that utilize shorter substrates, but it has a deeper active site to accommodate the longer 30-carbon substrate. An electron density map of the complex with a substrate analogue is presented in Figure S36, **but the map quality is poor and the fit of the model is not convincing. Moreover, it appears that not all of the olefin bonds are trans in the blue conformer here and also in Figure 2d (the fourth isoprenoid group appears to be cis; this should be a trans C=C bond).**

The cryo-EM structure of full-length MpMS reveals density for a hexameric prenyltransferase, similar to that first observed for human geranylgeranyl diphosphate synthase [Kavanagh et al. (2006) The crystal structure of human geranylgeranyl pyrophosphate synthase reveals a novel hexameric arrangement and inhibitory product binding. *J. Biol. Chem.* 281, 22004]. The cyclase domains of MpMS are disordered. As the authors note, this structure is similar to that of fusacoccadiene synthase (PaFS, ref. 33), which consists of a central prenyltransferase octamer or hexamer surrounded by randomly placed cyclase domains. Covalent crosslinking of MpMS with glutaraldehyde allowed for the identification of density corresponding to cyclase domains associated with the prenyltransferase hexamer (Figure 3). The cryo-EM structure of glutaraldehyde-crosslinked PaFS similarly revealed density for an associated cyclase domain which was confirmed using mass spectrometry (ref. 33).

However, association sites are on the "top" and "bottom" of the MpMS prenyltransferase hexamer, whereas association sites are on the "side" of the PaFS octamer. Differences in prenyltransferase-cyclase association represent an important finding. The manner by which these two domains associate leads to alternative models for understanding substrate channeling in catalysis, as discussed on lines 332-342. The authors conclude that this study "broadens the boundaries of class I TSs functions and enriches our understanding towards terpene biosynthesis in nature" (lines 444-445).

In summary, the most important aspect of this work is the discovery of class I triterpene synthases that generate and cyclize the 30-carbon substrate hexaprenyl diphosphate. This broadens the family of class I terpene cyclases, both prenyltransferases and cyclases. Detailed studies of the newly identified class I triterpene cyclases reveal familiar structures and mechanistic features as previously encountered in other class I enzymes. The biological importance of the newly discovered triterpenes is yet to be determined.

Response: We really appreciate the reviewer's support and instructive comments. Although we have tried to obtain the complex structure with 2,3-dihydro-HexPP using different soaking and co-crystallization methods for more than one year, we could not obtain better data set so far. It is difficult to improve the ligand occupancy over the current data. Therefore, the structure of TvTS-TC with 2,3-dihydro-HexPP was revised. Since the electron density of the prenyl moiety was too weak, the ligand was revised to PPI in the deposited structure (PDB: 7VTB). The new validation report is provided as supplementary file. The complex structure of TvTS-TC with 2,3-dihydro-HexPP is now shown as a fitting model in the manuscript, and the fourth double bond of the ligand in the purple conformation was revised to a *trans* double bond.

The descriptions of the structure in the main text have been modified as follows. "*Although the hydrocarbon chain of 2,3-dihydro-HexPP was not clearly observed, the disordered regions in the apo structure, especially the DDXXD motif and the active site loop D173-D182, appeared clearly structured in the complex, indicating a major conformational change upon substrate surrogate binding that facilitates active site closure (Figs. 1a, Extended Data Fig. 1c). Modelling of the 2,3-dihydro-HexPP electron density suggested two possible conformers (Figs. 2a and S35), one stretched out across the active site (yellow) and the other prefolded (purple) for C1-III-IV cyclization with C1-C11 and C10-C14 distances of 5.4 and 3.2 Å, respectively.*" (page 5, lines 135-142) Additionally, Figure 2, Extended Data Figure 1 and 2, Supplementary Figure 36, and Table S7 have been modified.

Although the complex structure has been modified, the cyclization mechanism of TvTS was supported by isotope experiments. Further, the crystal structures and mutagenesis study clearly demonstrated that the active site volume of TvTS is sufficiently large to accommodate Hex-PP. We thus believe the present results can stand alone in terms of substance and novelty as a communication of an important discovery.

Referee #2 (Remarks to the Author):

This is an excellent manuscript. The discovery of a family of alternative triterpene synthases that utilise a substrate other than squalene/2,3-oxidosqualene is a landmark advance. The body of work is substantial, impressive and very thorough. I have only a few questions/comments:

1)The authors chose ten chimeric TSs with low similarity to previous TSs and built their structures using AlphaFold2. Docking showed that five of these TSs have the potential to bind HexPP based on the finding that they had larger binding pockets. These five were tested experimentally for the production of triterpenes, of which one did indeed form a triterpene. **It would be good to include experimental data for the other five that were not chosen based on their AlphaFold2 model, as it would act as a validation of the AlphaFold2 method.**

Response: We thank the reviewer for giving this suggestion, we have verified the function of the other five candidates (PTTC035, 058, 059, 062 and 074) that were not chosen based on TvTS-based AlphaFold2 prediction model, and the resulting data showed that PTTC074 can produce tiny amount of terterpene (white asterisks), the other four enzymes were unable to produce a triterpene (Fig. R1). GCMS data showed that triterpene produced by PTTC074 with same retention time and mass fragment as compound 3 (colleterpenol) that produced by Cgl13955 (Fig. R2). However, low titer restricted the structure characterization of this compound.

Figure R1 (Extended Data Figure 7) | Product detection of triterpenes produced by chimeric class I triterpene synthase in *S. cerevisiae*. Engineered *S. cerevisiae* containing i) PTTC027, ii) PTTC035, iii) PTTC044, iv) PTTC058, v) PTTC059, vi) PTTC060, vii) PTTC062, viii) PTTC074, ix) PTTC114, x) *Cgl13855* overexpressed under the control of GAL10 promoter; xi) blank, engineered efficient terpene precursor providing chassis *S. cerevisiae* YZL141. Red asterisks indicate the peak for the triterpene **3**, white asterisks indicate the similar peak produced by PTTC074.

Figure R2 (Figure S44) | GCMS detection of **3 produced by a) *S. cerevisiae* PTTC074 and b) *S. cerevisiae* RC181.** *S. cerevisiae* RC181 harboring the *Cgl13855* (*CgCS*) from *C. gloeosporioides* ES026. White asterisks indicate the peak produced by PTTC074 with same retention time and mass fragment as compound **3** (colleterpenol) that produced by *CgCS*.

The AlphaFold2-based chimeric class I triterpene prediction model was developed based on the structure of TvTS, which represents a class of enzymes with deep binding pockets, the HexPP can bind with a relatively stretched conformation. PTTC074 cannot match this model, so it was excluded. Structure prediction and docking data showed that MpMS represents another type of enzyme with a wide and shallow binding pocket. Because of steric hindrance caused by the residue M321 at the bottom of the MpMS binding pocket, HexPP can only bind with a folded conformation (Fig. S42 b and f). The binding pocket of PTTC074 shows similar feature with this model (Fig. R3, Fig. S43 k and l), so we infer that this is the reason why PTTC074 can also produce a triterpene (compound 3). This data enriched and refined our AlphaFold2-based class I triterpene synthase prediction model, and implying the exist of more triterpene synthases to be explored.

We have added these data in the revised manuscript in lines 225-227, Extend Data Fig. 7, Fig. S43 k and l and Fig. S44.

Figure R3 (Figure S43i-l) | AlphaFold2-based structure prediction and the substrate docking of class I triterpene candidates. AlphaFold2 predicted substrate binding pocket docking with HexPP of a) Cgl13855-TC and c) PTTC074-TC. b) The alignment of binding pockets between TvTS-TC (green sticks, showed as surface) and Cgl13855-TC (yellow sticks), d) The alignment of binding pockets between MpMS (green sticks, showed as surface) and PTTC074-TC (magenta sticks). Green sticks in a and c represent residues of receptors, cyan sticks represent ligands and purple balls represent Mg^{2+} .

2) Perhaps beyond the scope of this paper, but it would be interesting to investigate the expression of these genes at different developmental stages/under different environmental conditions (e.g. minimal nutrients v nutrient rich media) to try to gain some insight into when these novel triterpenes may be produced.

Response: We selected six different media (TY, YM, Peter, FM1, FM4, ISP4) to ferment and culture the host fungi of TvTS and CgCS, and the fermentation results were analyzed by HPLC, which showed that different media had quite a significant effect on the secondary metabolites of these two strains. However, based on the characteristic absorption peaks of terpenoids, we did not find significant change for terpenoid production in the different media of these two strains. It is possible that the laboratory culture conditions are unable to meet the needs of TvTS, CgCS and related genes' expression in *Talaromyces verruculosus* (Fig. R4) and *Colletotrichum gloeosporioides* (Fig. R5), and maybe result in silent genes. We are planning to characterize the derived triterpenoids in our newly developed efficient terpenoid precursor (IPP and DMAPP) providing *Aspergillus oryzae* chassis in the near future.

Since this part is beyond the scope of this article, we just answer this question here and did not put this data in the article.

Media used in this study: **Peter** (Glucose 20 g, Yeast extract 3 g, Malt extract 3 g, Peptone 5 g, Water 1000 mL, pH 7.4). **YM** (Yeast extract 3 g, Malt extract 3 g, Peptone 2 g, Water 1000 mL, pH 7.2). **FM1** (Malt extract 3 g, Glucose 20 g, KH_2PO_4 3 g Water 1000 mL, pH 6.5). **FM4** (Soluble starch 10 g, Yeast extract 2 g, Peptone 2 g, KH_2PO_4 1 g, K_2HPO_4 3 g, MgSO_4 2 g, Water 1000 mL, pH 7.3). **ISP4** (Soluble starch 10 g, K_2HPO_4 1.0g, $\text{MgSO}_4 \cdot 7\text{H}_2\text{O}$ 1.0g, NaCl 1.0g, $(\text{NH}_4)_2\text{SO}_4$ 2.0g, CaCO_3 2.0g, $\text{FeSO}_4 \cdot 7\text{H}_2\text{O}$ 0.001g, $\text{MnCl}_2 \cdot 7\text{H}_2\text{O}$ 0.001g, Water 1000 mL, pH 7.4). **TY**: (Peptone 10 g, Yeast extract 3 g, Glucose 20 g, NaCl 5.0g, Water 1000 mL, pH 7.3).

Figure

R4. Detection of natural products produced by *Talaromyces verruculosus* TS63-9 in different culture medium.

Figure R5. Detection of natural products produced by *Colletotrichum gloeosporioides* ES026 in different culture medium.

3) Is there any evidence that related enzymes might exist in plants?

Response: We thank the reviewer for raising this interesting question. We carry out this experiment to address your concern. We collected the sequences of 12 characterized sesterterpene synthases from plants which were used as query sequences to BLAST plant protein sequences in the NCBI database. The resulting sequences were then used to remove duplicates, filtered with sequence length lower than 650 residues, clustered by CD-HIT with sequence identity cut-off 0.7. Finally, 31 candidate sequences derived from plants were obtained including 12 query sequences. Except the reported sesterterpene synthase AtTPS18 of which a crystal structure is known, the structures of the other 30 candidates were predicted using AlphaFold2. We analyzed the AtTPS18 crystal structure (PDB ID: 7BZC) and the results (**Fig. R6a**) showed the depth of AtTPS18 binding site was about 18.4 Å, and two residues, Y412 and M526, at the bottom might hinder the enzyme to bind ligands with long carbon chains such as HexPP, because of the steric hindrance caused by the large side chains of Y412 and M526. The predicted structures were then aligned to the crystal structure of AtTPS18 and the binding site residues were compared. Except candidate ***Redacted*** the binding sites of other 29 candidates showed similar features like AtTPS18, so we inferred these candidates may not produce the triterpenoid like we reported in this study. In addition, we docked HexPP to TPS06 (**Fig. R6b**), the result showed that the depth of TPS06 binding site was about 23.6 Å. Because the bottom residue, AtTPS18 Y412, was converted to TPS06 A431, which had a smaller side chain and reduced the steric hindrance, leading to possibly bind HexPP. We speculate that it is possible for TPS06 to use HexPP as substrate to produce triterpene compound.

Since this part is beyond the scope of this article, we just answer this question here and did not put this data in the article.

Figure R6. The binding site of AtTPS18 and TPS06. a) The surface of binding site of AtTPS18 (PDB ID: 7BZC); b) The binding site of TPS06 docking with HexPP.

Reviewer Reports on the First Revision:

Referees' comments:

Referee #1 (Remarks to the Author):

The revised manuscript is improved, but some issues with the crystallography remain. Due to the poor fit of the substrate surrogate in the electron density map (Fig. S36), the authors revised the bound ligand to PPI in the structure deposited in the PDB. However, electron density even just for PPI is not convincing, with oxygen atoms of one of the phosphate groups sticking out of density (see Fig. S36; also see last page of PDB validation report). Regardless, the authors do not mention in the text that they refined this density as PPI. This should be noted somewhere on page 5, lines 136-143.

At 2.0 Å resolution, the density for isoprenoids should be better, but the authors note in their rebuttal that they "could not obtain better data set so far". If you look closely at Fig. S36, each isoprenoid methyl group sticks out of density. The tubular density looks more consistent with a molecule of PEG (used in the crystallization buffer) rather than the substrate surrogate. Can the authors make a convincing argument that this density is not PEG? If not, this possibility should be noted somewhere on page 5, lines 136-143.

It is puzzling why the authors still present the previously refined model of the enzyme-substrate surrogate complex in Figs. 2a and S36 and Extended Data Figs. 1c and 2a, with no mention of the fact that this model does not derive from refined atomic coordinates in the current crystallographic model. This is confusing.

Referee #2 (Remarks to the Author):

The authors have addressed all of my comments and revised the manuscript accordingly.

Author Rebuttals to First Revision:

We thank reviewers for their positive recommendation and valuable suggestions! We have carefully considered all the comments and provided our responses in a point-by-point format.

Reviewer #1

The revised manuscript is improved, but some issues with the crystallography remain. Due to the poor fit of the substrate surrogate in the electron density map (Fig. S36), the authors revised the bound ligand to PPi in the structure deposited in the PDB. However, electron density even just for PPi is not convincing, with oxygen atoms of one of the phosphate groups sticking out of density (see Fig. S36; also see last page of PDB validation report). Regardless, the authors do not mention in the text that they refined this density as PPi. This should be noted somewhere on page 5, lines 136-143.

At 2.0 Å resolution, the density for isoprenoids should be better, but the authors note in their rebuttal that they "could not obtain better data set so far". If you look closely at Fig. S36, each isoprenoid methyl group sticks out of density. The tubular density looks more consistent with a molecule of PEG (used in the crystallization buffer) rather than the substrate surrogate. Can the authors make a convincing argument that this density is not PEG? If not, this possibility should be noted somewhere on page 5, lines 136-143.

We really appreciate the reviewer's critical and thoughtful comments. According to the suggestion, we re-examined the density map very carefully, and decided not to include PPi in the deposited structure. Please see attached the revised validation report and the detailed explanation below.

Following the reviewer's suggestion, we thus tested fitting the model of 2,3-dihydro-HexPP or PEG into the strong extra density observed close to the DDXXD motif and in the active site cavity after soaking with 2,3-dihydro-HexPP (Fig. S36a and b). Firstly, we added the model of 2,3-dihydro-HexPP in the density, but some of isoprenoid methyl groups stick out of density (Fig. S36c). Then, we added the model of PEG in the density. However, the refined structure still contains large unassigned densities close to DDXXD motif (Fig. S36d). These observations suggested that it is still possible that

the density contains both 2,3-dihydro-HexPP and PEG but with low occupancies. However, since we agree that the electron density even just for PPI is not very convincing, we decided not to include the ligand in the deposited structure. Instead, we constructed the docking model of 2,3-dihydro-HexPP based on the observed density, which is now clearly explained in the revised manuscript and in Fig. S36e. Although we cannot conclude with certainty that the observed electron density is that of 2,3-dihydro-HexPP, the disordered regions in the apo structure, especially the DDXXD motif and the active site loop D173-D182 (shown in salmon), appeared clearly structured by the soaking, suggesting significant conformational changes upon substrate binding that facilitates active site closure (Extended Data Fig. 1a, c). Moreover, the results from our mutagenesis studies are in line with the model obtained by docking of HexPP in the active site of TvTS (Extended Data Fig. 2).

Figure S36 | Additional electron density observed after soaking of TvTS with the non-reactive substrate surrogate 2,3-dihydro-HexPP. Electron density map of **a**, TvTS (apo) and **b**, TvTS after soaking with 2,3-dihydro-HexPP. The additional electron density observed close to the DDXXD motif is likely from 2,3-dihydro-HexPP, but could not be assigned with certainty. The 2mFo-DFc (blue mesh) and the mFo-DFc maps (positive: green mesh, negative: red mesh) are represented and contoured at 1.2 and 3.5 σ , respectively. **c**, Fitting of 2,3-dihydro-HexPP reveals that some of isoprenoid methyl groups stick out, while for **d**, fitting of polyethylene glycol (PEG) as used in the crystallization buffer density close to DDXXD motif remained unassigned. The 2mFo-DFc and mFo-DFc maps are represented and contoured at 0.8 and 3.5 σ , respectively. Since in **c** and **d** both possible ligands did not fit perfectly, no modelled ligand was added to the deposited structure

(7VTB). **e**, For further structure based experimental work, docking of 2,3-dihydro-HexPP in two possible conformations to the TvTS active site was performed. The Fo-Fc polder omit map of ligands are represented as a gray mesh, contoured at $+3.0 \sigma$.

Extended Data Figure 1 | The structure of the cyclase domain of TvTS. **a**, The apo-form of TvTS-TC (green). The DDXXD, NSE, and RY motifs (red box) are conserved, and the active site forming regions, including aspartic-rich metal binding DDXXD motif, the region (V234-V243) after the NSE motif, and the A156-C183 region are disordered; **b**, PaFS-TC in complex with neridronate (cyan cartoon with magenta sticks, PDB: 5ER832, in comparison to TvTS-TS: rmsd values of 1.6 \AA for C α -atoms, 47% amino acid sequence identity); **c**, partially closed conformation of TvTS-TC. The disordered regions in the apo structure, especially the DDXXD motif and the active site loop D173-D182 (shown in salmon), appear clearly structured after soaking with 2,3-dihydro-HexPP. The docking model of 2,3-dihydro-HexPP was constructed based on the observed density and two possible conformers are shown by yellow and purple sticks.

We have revised the manuscript as follows:

Page 5, line 138-147,

“Although 2,3-dihydro-HexPP was not clearly observed, and it cannot completely be excluded that the observed electron density originates from polyethylene glycol used in the crystallization buffer (Fig. S36), the disordered regions in the apo structure, especially the DDXXD motif and the active site loop D173-D182, appeared clearly structured upon soaking with the substrate surrogate. This observation suggests a significant conformational change of TvTS upon substrate binding that facilitates active site closure (Extended Data Figs. 1a and c). The docking model of 2,3-dihydro-HexPP based on the observed electron density suggested two possible conformers (Figs. 2a and S36e), one stretched out across the active site (yellow) and the other prefolded (purple) for C1-III-IV cyclization with C1-C11 and C10-C14 distances of 5.4 and 3.2 \AA , respectively.”

Method section, page 20, line 607-619,

“After soaking with 2,3-dihydro-HexPP, strong additional electron densities were observed close to the DDXXD motif and in active site cavity (Fig. S36a and b). Modelling of 2,3-dihydro-HexPP to the observed density was partly satisfying, but some of the methyl groups along the isoprenoid chain stucked out (Fig. S36c). For an alternative explanation, PEG used in the crystallization buffer was modelled to the density, but in this case in the refined structure large unassigned densities close to DDXXD motif remained (Fig. S36d). Thus it is possible that the observed density originates from both 2,3-dihydro-HexPP and PEG with low occupancies. The final crystal data and intensity statistics are summarized in Table S7. The Ramachandran statistics are as follows: 97.6% favored, 2.4% allowed for TvTS-TC apo, 98.9% favored, 1.1% allowed for TvTS-TC soaked with 2,3-dihydro-HexPP. Although the ligand was not assigned, the conformation of PEG and 2,3-dihydro-HexPP in the active site should be similarly defined by active site residues. Therefore, a docking model of 2,3-dihydro-HexPP based on the observed density was developed.”

It is puzzling why the authors still present the previously refined model of the enzyme-substrate surrogate complex in Figs. 2a and S36 and Extended Data Figs. 1c and 2a, with no mention of the fact that this model does not derive from refined atomic coordinates in the current crystallographic model. This is confusing.

We apologize for the confusion. According to the reviewer’s suggestion, we have clearly explained that this is the “docking model” of 2,3-dihydro-HexPP in the legend of Figs. 2a, S36, and Extended Data Figs. 1c and 2a.

Reviewer #2:

The authors have addressed all of my comments and revised the manuscript accordingly.

We appreciate the reviewer’s comment.

We believe the present results can stand alone in terms of substance and novelty as a communication of an important discovery. The most valuable finding in our research is discovery and functionally characterization of non-squalene triterpenes. The proposed cyclization mechanism of TvTS was supported by isotope experiments. Furthermore, the crystal structures, cryo-EM structures, docking model, and mutagenesis studies clearly demonstrated that the active site volume of TvTS and MpMS is sufficiently large enough to accommodate HexPP. We hope that our revisions and responses will meet the high standards of the reviewers.

The class I triterpene synthases represent a unique class of triterpene synthase, and we believe that they are ubiquitously distributed in nature. We look forward to sharing this important discovery with readers as early as possible.

Reviewer Reports on the Second Revision:

Referees' comments:

Referee #1 (Remarks to the Author):

The re-revised manuscript is outstanding. The crystallographic results are now cautiously interpreted with regard to the extra density in the active site. I like the author's interpretation that it might be a mixture of substrate analogue and PEG. This does not make the density useless - it still provides a valuable guide for modeling the bound substrate conformation. Both PEG and a long isoprenoid are flexible molecules and it is not unreasonable to expect that they might bind with similar conformations in a largely hydrophobic active site cavity. Thus, it is very much ok to leave this density without a molecular model fit; indeed, it reflects a careful approach with the crystallography. No further revision is necessary, I look forward to seeing this paper in print.